**www.cambridge.org/ext**

# After the mammoths: The ecological legacy of late Pleistocene megafauna extinctions

Felisa A. Smith[1] ⬤, Emma A. Elliott Smith[2], Carson P. Hedberg[1], S. Kathleen Lyons[3], Melissa I. Pardi[4] ⬤ and Catalina P. Tomé[5]

[1]Department of Biology, University of New Mexico, Albuquerque, NM, USA; [2]Department of Anthropology, United States National Museum of Natural History, Washington, DC, USA; [3]School of Biological Sciences, University of Nebraska-Lincoln, Lincoln, NE, USA; [4]Research and Collections Center, Illinois State Museum, Springfield, IL, USA and [5]Indiana State Museum and Historic Sites, Indianapolis, IN, USA

## Review

**Keywords:**
biodiversity; climate change; ecosystem engineer; Hall's cave; large-bodied mammals; stable isotopes

**Author for correspondence:**
Felisa Smith,
Email: fasmith@unm.edu

### Abstract

The significant extinctions in Earth history have largely been unpredictable in terms of what species perish and what traits make species susceptible. The extinctions occurring during the late Pleistocene are unusual in this regard, because they were strongly size-selective and targeted exclusively large-bodied animals (i.e., megafauna, >1 ton) and disproportionately, large-bodied herbivores. Because these animals are also at particular risk today, the aftermath of the late Pleistocene extinctions can provide insights into how the loss or decline of contemporary large-bodied animals may influence ecosystems. Here, we review the ecological consequences of the late Pleistocene extinctions on major aspects of the environment, on communities and ecosystems, as well as on the diet, distribution and behavior of surviving mammals. We find the consequences of the loss of megafauna were pervasive and left legacies detectable in all parts of the Earth system. Furthermore, we find that the ecological roles that extinct and modern megafauna play in the Earth system are not replicated by smaller-bodied animals. Our review highlights the important perspectives that paleoecology can provide for modern conservation efforts.

### Impact statement

Our review discusses the ecological consequences of the late Pleistocene extinctions on major aspects of Earth systems as well as on the diet, distribution and behavior of surviving mammals. We demonstrate that the late Pleistocene loss of megafauna was pervasive and left legacies detectable within the modern atmosphere, geosphere, hydrosphere and biosphere. Moreover, the ecological roles that extinct and modern megafauna play in the Earth system are not replicated by smaller-bodied animals. Our review highlights the importance of integrating a paleontological perspective into modern conservation efforts to develop a more synoptic understanding of ecosystems.

### Introduction

Paleontologists are fond of noting that extinction is the ultimate fate of any species. After all, turnover is a natural ongoing process in ecological communities; the average duration of a species is about 1–3 Ma (Raup, 1981, 1991; Foote and Raup, 1996; Alroy, 2000; Vrba and DeGusta, 2004). Only rarely does the rate of extinction within a short temporal window rise to the level where we consider it unusual – a mass extinction event. There have been five of these that we know of in the geologic record (Raup and Sepkoski, 1982; Sepkoski, 1984), and an additional widespread event during the late Pleistocene/Holocene (~100 ka to modern) that some modern scientists refer to as a mass extinction (May et al., 1995; Pimm et al., 1995; Kolbert, 2014; Ceballos et al., 2015, 2017). This latter event was unusual in a number of ways.

First, while certainly catastrophic, extinctions during the late Quaternary do not (or have not yet) risen to the level of a mass extinction. They fell well short of the threshold of 75% of species loss and moreover, were taxonomically selective targeting largely mammals and nonvolant birds (Martin and Klein, 1984). Only if we project into the future and consider ongoing biodiversity loss part of the same process do we approach the threshold where this event rises to a mass extinction (Barnosky et al., 2011; Plotnick et al., 2016; Smith, 2021). Second, the late Pleistocene extinctions were both spatially and temporally transgressive. They began earlier in Eurasia (~100–80 ka), then spread to Australia (~60–50 ka), North and South America (15–12 ka; Barnosky et al., 2004; Lyons et al., 2004; Smith et al., 2018, 2019) and finally islands (Holocene; Turvey, 2009; Slavenko et al., 2016; Fromm et al., 2021). Furthermore, the intensity of extinctions was only slightly elevated from background in Eurasia and Africa (Smith et al., 2010). Third, these extinctions were

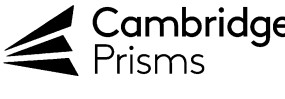



strikingly size- and diet-biased, disproportionately targeting large-bodied herbivorous mammals (Lyons et al., 2004; Smith et al., 2018). Finally, there was a clear and novel mechanism for this extinction – humans (Martin, 1967, 1984, 2005; Martin and Steadman, 1999; Roberts et al., 2001; Lyons et al., 2004, 2016b; Miller et al., 2005; Koch and Barnosky, 2006; Haynes, 2009; Turvey, 2009; Turvey and Fritz, 2011; Sandom et al., 2014; Smith et al., 2018, 2019).

Interestingly, while considerable research has focused on the role of humans in the terminal Pleistocene megafauna extinctions (e.g., Martin, 1967, 1984, 2005; Lyons et al., 2004, 2016a; Miller et al., 2005; Koch and Barnosky, 2006; Sandom et al., 2014), much less has focused on the *environmental legacy* of the loss of tens of millions of large-bodied individuals from the landscape (but see Johnson, 2009; Malhi et al., 2016; Smith et al., 2016a, 2019; Galetti et al., 2018; Smith, 2021). Given the precarious state of large-bodied mammals today (Cardillo et al., 2005; Dirzo et al., 2014; Ceballos et al., 2015, 2017; Ripple et al., 2014, 2019), it may well be that more than any other extinction in the history of life on our planet, the late Pleistocene megafauna extinction can inform modern conservation biology by providing insights about what might happen if our current biodiversity crisis is not slowed or halted (Barnosky et al., 2016; Malhi et al., 2016; Smith et al., 2016a, 2018; Galetti et al., 2018; Smith, 2021).

To assess the influence of the terminal Pleistocene extinction on Earth systems, we first need to appreciate just how uniquely size biased it was; no other extinctions in the Cenozoic mammal record approached the degree of size selectivity seen at this time (Alroy, 1999; Smith et al., 2018). Indeed, body mass is not usually a good predictor of extinction risk for mammals in the fossil record (Smith et al., 2018). However, by the late Pleistocene, there was a highly significant positive association between body size and extinction risk – the average mass of victims was more than 2–3 orders of magnitude higher than survivors (Smith et al., 2018). Moreover, this size bias occurred on each continent and within each trophic guild, and it followed the movement of humans across the globe (Lyons et al., 2004; Smith et al., 2018, 2019). Thus, in examining the legacy of this extinction, we are necessarily focusing on the consequences of truncating the right side of the body size distribution of mammals; that is, losing all of the largest bodied species within communities (Figure 1). In North America, for example, all mammals >600 kg went extinct (Lyons et al., 2004).

Large-bodied mammals play an important, even 'oversized' role in ecosystems (Owen-Smith, 1988). This arises from their large mass and high energetic requirements, which require physical space, abundant forage, large home ranges and dictate large-scale movement (Peters, 1983; Owen-Smith, 1988). Thus, megafauna potentially influence all aspects of the Earth system – the atmosphere, geosphere, hydrosphere and biosphere, leading to unique and pervasive influences not replicated by smaller-bodied animals (Hyvarinen et al., 2021). Logically, the megafauna of the terminal Pleistocene had a much greater impact on Earth systems because of their even larger size (Figure 2; Haynes, 2012; Smith et al., 2016a). The Columbian mammoth (*Mammuthus columbi*), for example, was approximately three times more massive than the average modern elephant and more than an order of magnitude larger than a domestic cow (Smith et al., 2003). Even carnivores were bigger; species of extinct sabertooth cats weighed up to 400 kg, about twice the average size of an African lion (Smith et al., 2003). Because many life history and ecological attributes of animals scale non-linearly with size (Peters, 1983), this suggests Pleistocene megafauna had a disproportionately greater influence on environments and biotic interactions than do modern megafauna (Smith et al., 2016a).

Here, we explore how essential activities, such as foraging, digestion and movement of megafauna influenced various physical aspects of the environment, vegetation structure and composition, nutrient cycling and gas exchange (Figure 2). Much of this work borrows from what we know about contemporary mammals and/or is based on modeling efforts. We also explore what we know thus far about the response of surviving mammals in the early Holocene to the extinction of their congeners, particularly in terms of potential dietary shifts, changes in morphology and/or behavior and distribution shifts. Finally, we examine how these changes influenced the overall function of communities at various spatial scales and the ecological interactions within them. Because much of our collective expertise is with North American ecosystems, we often use this continent as a case example. Moreover, many of our more detailed explorations are taken from work at Hall's Cave, an exceptional well-dated late Quaternary fossil site in the Edward's Plateau of Texas where we have worked for the past 8 years. However, we note our findings are likely applicable globally, and when and where we can, we highlight research in other ecosystems and continents. Our aim is to provide a synoptic understanding of the ecological legacy of the terminal Pleistocene megafaunal extinction on surviving animals and ecosystems.

## Movement

Being very large brings physical challenges as animals interact with their environment (Owen-Smith, 1988). Here, we discuss how megafauna influence the landscape through their movements up and down slope, over large distances, and the consequences of their mass on soil and water tables. While researchers have explored these effects to some extent with extant animals (i.e., Owen-Smith, 1988), extinct megaherbivores almost certainly had much more profound influences on their environment; these remain poorly characterized.

### *Slope*

Body size influences movement (Peters, 1983). For example, the path that a large-bodied mammal takes as it travels up and down mountain slopes is different than that of smaller mammals

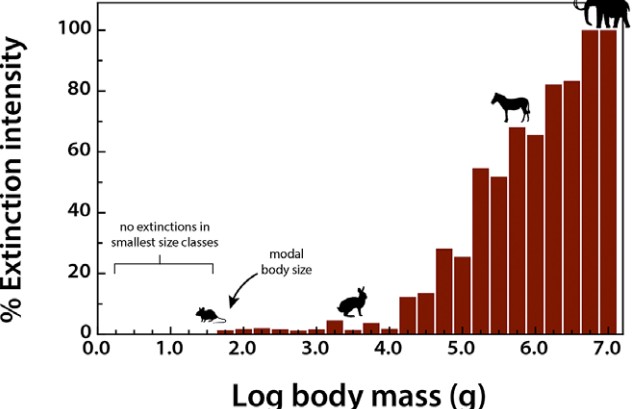

**Figure 1.** Proportion of extinctions within each body size class of terrestrial nonvolant mammals over the late Pleistocene and Holocene. Data are averaged across the globe; insular mammals are not shown. Because the median body mass of mammals is under 100 g ($log_2$), most mammals were unaffected by this event; however, extinction risk rose sharply with larger body mass. Such extreme size-selectivity is unique in the vertebrate fossil record (Data from Smith et al., 2018).

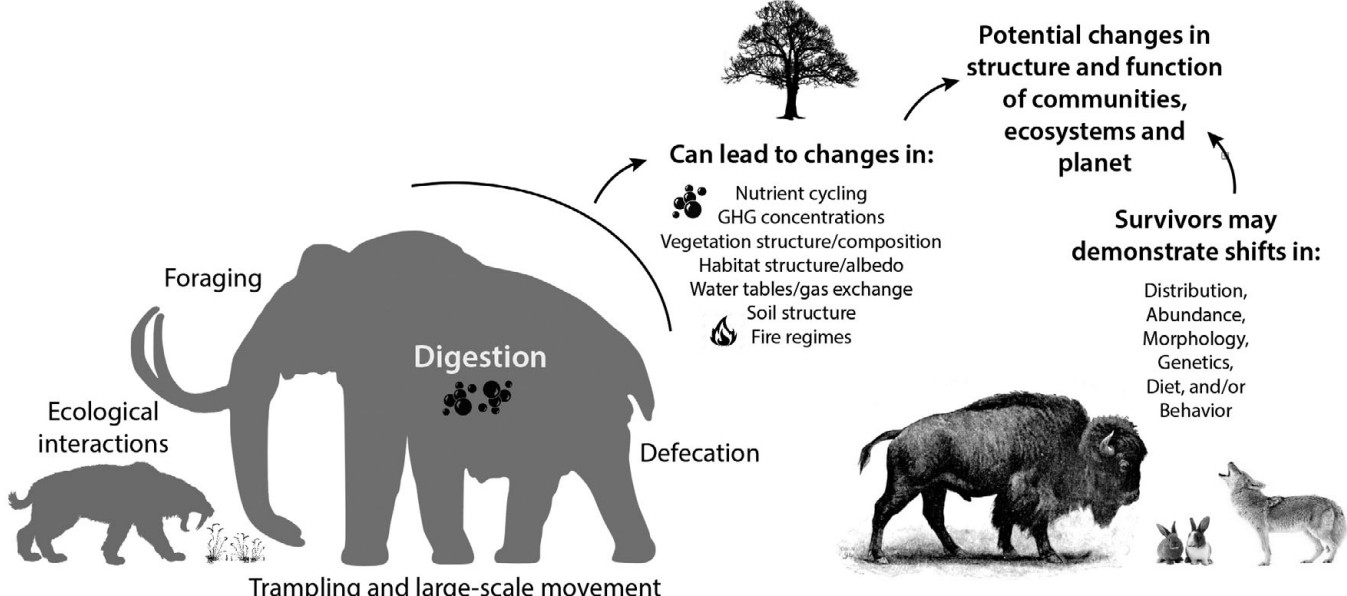

**Figure 2.** Consequences of the loss of large-bodied mammals in terrestrial ecosystems. Megafauna were (and are) unique members of ecological communities and their physiological, life history, and ecological activities have disproportionate influences on community, ecosystem and earth system processes. Silhouettes here and elsewhere from Phylopic (http://phylopic.org).

(Reichman and Aitchison, 1981; Figure 3). This reflects substantial differences in the energetic costs of transport, which are mediated by body mass, gravity and the slope of the incline. For example, for a horse to move up a 15% incline requires a 630% increase in metabolism relative to moving along a flat surface (Reichman and Aitchison, 1981). Consequently, trail angles significantly decrease with increasing body size (Figure 3). While small mammals may scamper straight up a slope, larger ones make many turns

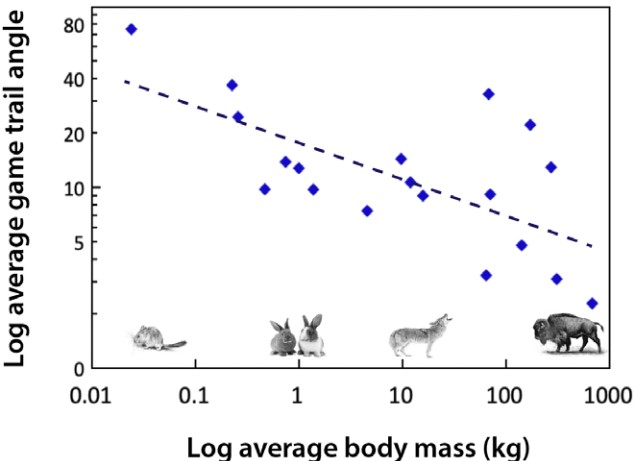

**Figure 3.** The angle of animal game trails up mountain slopes as a function of body mass. Data were extracted from Figure 2 in Reichman and Aitchison (1981) and are binned by species; mammals plotted range from a mouse (0.025 kg) to a bison (680 kg). Trail angles were determined by measuring footprints in snow and include measurements conducted on a variety of different slope inclines. Since the trail angle is influenced by slope incline (Reichman and Aitchison, 1981) this contributes to considerable scatter in the data. If data had been standardized for slope, this relationship might be steeper. Equation: Log trail angle (degrees) = −0.1935 log body mass (kg) + 1.2415; df = 17; $r^2$ = 0.4158; $P < .01$.

as they traverse a mountain. It follows that the difference in the way animals walk up (or down) a mountain slope leads to differences in the steepness and physiographic terracing of game trails when megaherbivores are present. The permeance of such trails is also influenced by the weight and likely the foot type of the animal; megaherbivores with hard hoofs are much more likely to produce hard-packed trails. There are likely environmental consequences; studies of hiking trails have demonstrated that modification of slopes can lead to erosion, changes in water movement if the trail channels rain water or snow melt, and influence surrounding vegetative communities (Yoda and Watanabe, 2000). While national parks and other entities have examined the physiographic effects of hiking trails (e.g., Yoda and Watanabe, 2000), little work to date has examined how game trails made by animals of different body sizes might influence the landscape.

### Distance

Modern megafauna move enormous distances (Owen-Smith, 1988). Even within their greatly restricted modern geographic range (e.g., Mahmood et al., 2021), elephants can travel more than 2,800 km annually (Mills et al., 2018). Yet, this is only a fraction of their historic movement patterns. In the past, mammoths and other large-bodied mammals may have roamed across entire or even multiple continents (Kahlke, 2015; Wooller et al., 2021; Miller et al., 2022). While we know little about landscape effects of mammoth movement, modern studies of elephants suggest as these giants move through the landscape they have detectable impacts on the density of trees and forest structure (Asner and Levick, 2012). Their bulk means that it is inevitable that they will knock down trees and create gaps in the vegetation as they move through forests. By creating gaps in what would otherwise be closed-canopy vegetation, megafauna provide opportunities for other plants to establish (Asner and Levick, 2012). The subsequent changes in forest

structure can be profound; elephants are two times more important than fire in regulating tree falls and recruitment (Asner et al., 2016). For example, elephant damage has led to a notable lack of small trees in Gabon's forest compared to the Amazon and reduced alpha diversity of the vegetation (Terborgh et al., 2016a, b). Not surprisingly, areas where megafauna are not present tend to demonstrate increases in woody cover (Asner et al., 2009, 2016; Gill et al., 2009; Asner and Levick, 2012; Barnosky et al., 2016). Given the relationship between geographic range size, home range size and body size (Brown, 1995) as well as the expected migration distances, extinct megafauna would have had effects across much larger areas than modern animals do today.

### Soil compaction

The largest extinct Pleistocene megafauna exceeded 10 tons (Smith et al., 2003) and even the smallest of them generally weighed more than a ton. Moreover, it is likely that many of these species traveled in social herds, much like modern large-bodied mammals. The concentration of large-bodied animals within an area undoubtedly led to localized compression of soils, which may have reduced habitat for burrowing animals such as gophers. Moreover, by removing air pockets, soil compaction can alter water and gas exchange with the atmosphere, which in turn, can have substantial regional and even global effects on climate (see section 'Biogeochemical change'). Soil compaction also influences permeability, which directly influences nutrient exchange and the diversity of microbes within soils (Wang et al., 2019). Several studies demonstrate that bioturbation and soil compaction by modern herbivores alters microbial composition (Liu et al., 2015; Wang et al., 2019); for example, regions where muskoxen graze in Greenland have much higher microbial diversity than do un-grazed soils (Aggerbeck et al., 2022). Others report that soil compaction and grazing by megaherbivores reduce carbon and nitrogen pools (Mosbacher et al., 2019).

### Foraging

Because megaherbivores rely on abundant, but low-quality resources, they necessarily spend the majority of their time foraging. The high absolute energetic requirements of megaherbivores, coupled with their large-scale movements, means their impacts are widespread and would have been even more so in the past. Thus, foraging has important ecosystem consequences, which we discuss below. These include changes in vegetation structure and composition, the openness and patchiness of vegetation, influence on fuel loads and fire regimes, long-distance dispersal of seeds, as well as physically disturbing the soil surface, changing the albedo of the surface, and recycling nutrients. Such effects vary along rainfall and soil fertility gradients (Augustine and McNaughton, 2006; Waldram et al., 2008; Pringle et al., 2016; Hyvarinen et al., 2021). Note that much of our speculation about the role of megaherbivores in Pleistocene ecosystems is based on studies of one system – modern African elephants in savannah ecosystems (Hyvarinen et al., 2021), which may provide slightly distorted inferences. However, it is likely that Pleistocene megafauna had somewhat similar, albeit much greater, impacts on late Quaternary ecosystems.

### Vegetation structure and composition

Numerous studies have documented the important influence of large-bodied herbivores on plant structure and composition, and on maintaining habitat heterogeneity (e.g., Owen-Smith, 1987,

1988, 1989; Cumming et al., 1997; Whyte et al., 2003; Western and Maitumo, 2004; Waldram et al., 2008; Johnson, 2009; Haynes, 2012; Rule et al., 2012; Keesing and Young, 2014; Asner et al., 2016; Bakker et al., 2016a, b; Ratajczak et al., 2022). Browsers, such as deer or antelope, tend to have an inhibiting effect on woody vegetation, while the effects of grazers, such as zebra, bison and horses, appear to be mixed (Bakker et al., 2016a, b). Large-bodied grazers can help maintain open grasslands and inhibit woodland regeneration through the trampling and destruction of tree seedlings, but they apparently can also have a positive effect on woodlands by reducing potential competition between grasses and shrubs (Johnson, 2009; Bakker et al., 2016a, b). Complicating our understanding of how multiple species of megaherbivores in the late Pleistocene would have impacted plant communities is that the dietary preferences of some megaherbivores appear to have been variable across time and/or space. For example, stable isotope analysis revealed that within a single location, camels (*Camelops* sp.) in central Texas foraged across the entire $C_3$ and $C_4$ vegetation spectrum during the late Pleistocene (Smith et al., 2022). Individual camels within the population could be characterized as browsers, mixed feeders or grazers, suggesting a high degree of individual specialization. Other work finds similar results for *Palaeolama* (Franca et al., 2015).

The maintenance of open grassland ecosystems at the expense of woodlands is beneficial for other herbivores; by shifting carbon storage from where it was locked inside woody tissues to the leaves of grass, carbon becomes more freely accessible (Johnson, 2009). Interestingly, foraging by bison in large herds alters the nutrient quality of the plants upon which they feed (Geremia et al., 2019). By feeding intensively in a local patch over a period of weeks, bison stimulate plant growth, but delay plant maturation causing the plants to remain high-quality forage for longer periods of time. In effect, they engineer their own 'green wave'. During the late Pleistocene, when large herds of bison would have been more common, grazing would have altered the quality of graze in the ecosystems where they occurred, facilitating the coexistence of other large herbivores. It is possible that the increased quality of forage would have supported larger population sizes, although this has not been investigated.

Because extinct megafauna were so much larger than contemporary animals, the physiographic effects they had on landscapes were more transformative. For example, modeling suggests that as much as 29% of the savanna woody cover in the South American continent may have been lost following the megafauna extinction (Doughty et al., 2016a). Furthermore, the heterogeneity of the landscape may also have been altered. Not only were woody savannas of South America more open and grassy when megafauna were present, but also there were mosaics of different vegetation types (Doughty et al., 2016a). These likely reflected differential space use by megafauna relative to smaller-bodied mammals; perhaps areas they could not access because of their size (Doughty et al., 2016a). In Australia, the loss of megafauna led to a wholescale transformation of mixed rainforest into sclerophyll vegetation (Rule et al., 2012). The northern hemisphere saw the loss of the vast 'mammoth steppe' – the largest biome of the Pleistocene consisting of a diverse assemblage of grasses, forbs and sedges – which had been maintained at least partially through grazing and other activities by large-bodied mammoth, camelids and bison (Zimov et al., 1995, 2012; Guthrie, 2001; Johnson, 2009).

Similarly, in the Neotropics, the megafauna extinction left a significant imprint on the current plant trait and ecosystem biogeography (Dantas and Pausas, 2020). Not only did some ecosystems shift from grasslands to forest after the extinction, but also the

distribution of plant defense traits (i.e., spinescence, leaf size and wood density), which had been correlated with Pleistocene megaherbivore density and distribution, became decoupled (Dantas and Pausas, 2020). This led to many 'evolutionary anachronisms'; that is, taxa whose ecological traits and adaptations are now obsolete for contemporary ecosystems (Janzen and Martin, 1982; Barlow, 2002; Guimarães et al., 2008). Consequently, the modern distribution of anti-herbivore defense traits in the Neotropics reflects the historical distribution of extinct megafauna rather than that of extant animals (Dantas and Pausas, 2020).

## Species diversity

An ecosystem engineer is a species whose activities increase or create habitat in an environment (Jones et al., 1994). While the degree of the effect varies between ecosystems (marine vs. terrestrial) and environments (forests, deserts, vs. grasslands), ecosystem engineers tend to increase species richness within the habitat (Jones et al., 1994; Romero et al., 2015; Coggan et al., 2018). Indeed, modern ecosystem engineers can increase species richness in a community by 25% (Romero et al., 2015). Very large-bodied animals are especially important as engineers (Owen-Smith, 1988) – a niche largely lost from communities after the late Pleistocene megafauna extinction.

Bison, one of the few surviving large-bodied taxa in North America, demonstrate this clearly; their reintroduction into tall prairie grasslands more than doubled native plant species richness (Ratajczak et al., 2022). This effect came about for several reasons. First, because bison consumed the dominant plant species in the habitat, they facilitated increases in the abundance and presence of other plant taxa. Second, their physical disturbances of the environment, such as the production of 'bison wallows' (depressions in the soil) greatly increased habitat heterogeneity, thereby also increasing plant species diversity. While grazing by cattle also increased plant diversity in tall grass prairie habitats, the effect was much weaker than that of the larger-bodied bison (Ratajczak et al., 2022).

Large-bodied mammals strongly influence the architecture of the environment with ecological engineering effects dependent on the mammal and habitat (Estes et al., 2011; Young et al., 2015; Goheen et al., 2018). In some ecosystems, megaherbivores may maintain habitat for other taxa, while in others, they reduce it (Keesing, 1998, 2000; Estes et al., 2011; Young et al., 2015). Either has implications for smaller-bodied vertebrates and the ecosystem. For example, the experimental exclusion of large-bodied herbivores from sites in Kenya and East Africa led to increases in small mammal abundance likely because of increased food quality (Keesing, 1998, 2000; Young et al., 2015). However, the increased density of small-bodied mammals led in turn to sharp increases in the number of medium and small-sized predators (Keesing, 1998, 2000). Cover also matters. In shrublands and savannahs, megaherbivore grazing can decrease small mammal biodiversity if the reduction of cover leads to more exposure to predators (Moser and Witmer, 2000). However, by damaging trees and increasing the structural complexity of the habitat, megaherbivores can also increase the ability of taxa to avoid detection by predators (Pringle, 2008). All of these effects are influenced by the size of the animal, and so would likely have been much greater in the past.

## Seed dispersal

Many plants rely on animals for seed dispersal. Mammalian seed dispersers serve that role through multiple pathways including transporting seeds that get caught in their fur, and through foraging. However, the effectiveness of mammal-mediated seed dispersal is influenced by body size. While foraging by small-bodied mammals has detectable effects on the structure of forests (e.g., Guimarães et al., 2008; Campos-Arceiz and Blake, 2011; Bueno et al., 2013; Pires et al., 2018), they do not replicate the role megafauna play in plant dispersal (Janzen and Martin, 1982; Barlow, 2002). Smaller-bodied herbivores collect and cache seeds, but often discard the kernel while eating some of the flesh of the fruit (Barlow, 2002; Johnson, 2009). Even if they eat seeds, they do not transport them as far as larger bodied herbivores because of their smaller home range (Kelt and Van Vuren, 2001). Indeed, on average extinct megaherbivores dispersed seeds 10 times further than do contemporary smaller-bodied herbivores (Pires et al., 2018). Moreover, germination for many large-seeded plants requires passage through the guts of large-bodied herbivores; this process removes the pulp and mechanically and/or chemically abrades the seed (Traveset et al., 2008). Many seeds are too large for surviving small-bodied herbivores to effectively process. Such plants are another example of an evolutionary anachronism (see section 'Vegetation structure and composition'). Even in regions with extant megaherbivores, large-seeded plants that depend on them for dispersal are declining in abundance or exhibiting a contraction of their geographic range because of reductions in the abundance of megafauna (Janzen and Martin, 1982; Barlow, 2002; Doughty et al., 2016c). For example, 4.5% of tree species in the Congo require an animal the size of an elephant (or larger) to disperse their seeds (Beaune et al., 2013). These trees are experiencing population declines because of decreases in the abundance of forest elephants. Although there is as yet little evidence of plant extinctions as a result of seed dispersal limitations, declines in tree abundance have important implications for carbon flux and biomass in forests (Doughty et al., 2016a, c).

## Fire regimes

The role of herbivores in mediating the fire regime within landscapes is unequivocal (Owen-Smith, 1987, 1988; Flannery, 1994; Waldram et al., 2008; Johnson, 2009; Johnson et al., 2018; Rouet-Leduc et al., 2021 and references therein). As discussed above, foraging alters the abundance of flammable woody vegetation and/or grasses, and promotes habitat heterogeneity, which in turn can influence the frequency, intensity and extent of fire within ecosystems (Owen-Smith, 1987, 1988; Bakker et al., 2016a; Johnson et al., 2018). Moreover, in the absence of herbivores, organic matter can accumulate and enhance both the frequency and intensity of fires. Given these effects, promotion of grazing by contemporary herbivores has been proposed as a tool for reducing wildfire risk in some habitats (Johnson et al., 2018; Rouet-Leduc et al., 2021). This appears to be most effective when a mix of browsers and grazers is employed (Rouet-Leduc et al., 2021). Other studies demonstrate livestock grazing appears to reduce fire frequency in tropical ecosystems, likely because grazing inhibits tree recruitment (Bernardi et al., 2019).

Given the results from modern studies, it is not surprising that a fundamental change in fire regimes has been reported for the late Pleistocene at many fossil localities following the megafauna extinction (Burney et al., 2003; Bond and Keeley, 2005; Gill et al., 2009, 2012; Rule et al., 2012; Gill, 2014; Bakker et al., 2016a, b). For example, sedimentary charcoal records indicate a widespread increase in fire across the globe from the late Pleistocene through

the Holocene, although these authors largely attributed it to climate shifts (Daniau et al., 2012). The shift in vegetation from mixed rainforest to sclerophyll vegetation in Australia resulted in an increase in fire frequency (Flannery, 1994; Rule et al., 2012). Not only were fires more frequent, but also they were likely more intense (Flannery, 1994; Waldram et al., 2008), which can lead to feedback loops between herbivore densities and fires. Larger, more intense fires can change herbivore foraging behavior in a way that increases the chances of more large fires (Waldram et al., 2008). Similar shifts in vegetation and fire frequency were reported in North America (e.g., Gill et al., 2009).

## Digestion

Large-bodied herbivores require enormous quantities of food. For example, a wild adult elephant can feed up to 18 h a day, and consume as much as 200–300 kg of vegetation within that time frame (Sukumar et al., 2006; https://elephantconservation.org). To meet these immense energetic needs, megaherbivores depend on plant structural carbohydrates such as cellulose and related components, which are the most abundant organic compounds within terrestrial landscapes (Van Soest, 1982). However, vertebrates lack cellulolytic enzymes to digest plant fiber. Thus, all terrestrial multicellular herbivores have evolved fermentation chambers housing symbiotic microflora to exploit these abundant resources. The 'giant fermentation vats' megaherbivores employ to digest plant fiber have many ecological and biochemical consequences and can influence Earth systems at global scales. We outline some of these processes, including changes in biochemical cycling and gas exchange, as well as nutrient flow across the landscape.

## Biogeochemical change

A byproduct of the microbial fermentation of plant fiber in the rumen or cecum is methane, a particularly potent greenhouse gas (Van Soest, 1982; Clauss and Hummel, 2005). Methane production scales positively with body mass (Smith et al., 2010, 2015, 2016b), so larger animals produce more methane than do smaller ones. Moreover, the very largest herbivores, such as those present in the late Pleistocene, may be digestively less efficient and thus experience even greater methanogenesis (Clauss and Hummel, 2005). Because atmospheric methane concentrations were lower at the terminal Pleistocene (e.g., ~680 to 700 vs. ~1895 ppbv $CH_4$ today), enteric fermentation was a proportionately greater source to the global pool. Thus, it is probable that the widespread extinction of herbivorous megafauna in the late Pleistocene influenced atmospheric gas exchange and global climate.

Several authors have calculated the enteric production of methane by megafauna at the terminal Pleistocene. For example, using a series of allometric equations relating methane production, geographic range and population density with body mass, Smith and colleagues (Smith et al., 2010, 2016b) calculated the annual decrease in the methane source pool resulting from the extinction. They found that methane production by late Pleistocene mammals totaled ~139 Tg yr.$^{-1}$, similar to what livestock contribute to the global budget today (Figure 4A; Smith et al., 2016b). Importantly, the terminal Pleistocene extinction resulted in an annual loss to the global atmospheric pool of ~69.6 Tg of $CH_4$, a ~35% reduction of the overall tropospheric input at this time (Smith et al., 2016b). Similar results have been obtained by others. For example, using a bottom-up approach, Zimov and Zimov (2014) calculated that Pleistocene mammals may have contributed 120–170 Tg to the annual global methane budget. Their estimate was based on

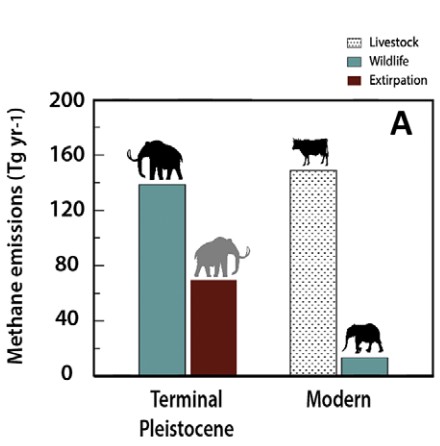

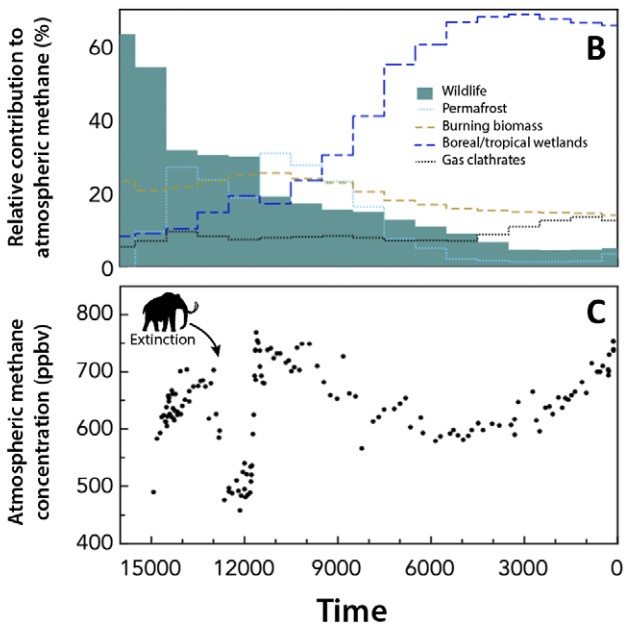

**Figure 4.** Changes in methane emissions by wildlife over time. (A) Enteric methane emissions by wild (teal) and domestic (spotted) herbivores. The reduction of $CH_4$ emissions resulting from extinctions or extirpation of animals is indicated in red. Shown are the emissions at the terminal Pleistocene and today. Modern emissions are largely sourced from domesticated livestock. Redrawn after Smith et al. (2016b). (B) The isotopic 'signature' of methane in ice cores. Drawn after data from Zimov and Zimov (2014). As wildlife has been extirpated across the globe, and more recently, has been replaced by domesticated livestock, there has been a corresponding shift in the sourcing of methane. Because methane is an important greenhouse gas, the amount and sourcing of emissions are important. (C) Methane concentrations from the GISP2 ice core (Brook, 2009). Note that there was an abrupt drop in global methane concentration in the atmosphere coincident with the extinction of megafauna at the terminal Pleistocene; this coincided with the onset of the Younger Dryas cold episode. Redrawn after Smith et al. (2016b).

computations of primary productivity, which they assumed was consumed by vertebrate herbivores.

This shift in the carbon biogeochemical cycle is evident in the ice core record (Figure 4B, C). Analysis of occluded air bubbles within the GISP2 and EPICA highly resolved ice cores (EPICA, 2004; Brook, 2009), reveals an abrupt drop in methane concentrations shortly after humans entered the New World, coincident with the extinction. This methane drop is significantly more abrupt (7–10 times) than other declines over the past 800 ka, suggesting novel contributing mechanisms (Figure 4C; Smith et al., 2016b). Moreover, there was a significant change in the sourcing of atmospheric methane. The isotopic value (i.e., $\delta^{13}C$ and $\delta^2H$) of methane varies with sources, meaning that it is possible to determine if a sample is derived from biomass burning, gas clathrates, permafrost, herbivores or boreal/tropical wetlands (Zimov and Zimov, 2014). While the Pleistocene isotopic signature suggests megaherbivores were the main contributors to the global pool, early Holocene methane emissions largely originated from the degrading frozen soils of the mammoth steppe biome (Figure 4B; Zimov and Zimov, 2014). Hence, it is highly probable these changes in atmospheric methane resulted from the widespread extinction of megafauna.

Interestingly, these decreases in methane concentration are also synchronous with the onset of the Younger Dryas stadial, an abrupt transition ~12.8 ka where climate returned to near glacial conditions in the higher latitudes of the Northern Hemisphere. Did the extirpation of hundreds of millions of large-bodied animals contribute in any way to this climatic event? Although methane is ~200 times less abundant than carbon dioxide in the atmosphere, its much greater efficiency in trapping radiation leads to a significant role in radiative forcing of climate. How much the decrease in methane contributed to the rapid drop in temperature at the Younger Dryas remains unclear, although simplistic modeling suggests a temperature drop of at least ~0.5°C (Smith et al., 2016b). The role of large vertebrates in regulating climate through the production of methane may not be confined to mammals; it has been speculated that herbivory by sauropods may have helped maintain warm Mesozoic climates (Wilkinson et al., 2012).

Note that paradoxically, the extinction of megaherbivores in the Americas may also have ultimately contributed to the rise in methane that occurred at the *end* of the Younger Dryas (Figure 4). As mentioned earlier, the wide-scale habitat alteration from the vast mammoth steppe of the Pleistocene to the more waterlogged habitats of the Holocene has been attributed in part to the absence of grazing by megaherbivores (Zimov et al., 1995; Guthrie, 2001; Johnson, 2009). Heavy grazing not only suppressed woody re-growth (Zimov et al., 1995) but may have also stimulated above-ground grass production and high rates of soil moisture transpiration (Zimov et al., 1995; Johnson, 2009). In the absence of heavy grazing by megaherbivores, water tables may have risen, leading to a slowdown in the rate of nutrient breakdown and recycling, an increase in organic matter accumulation, and a decrease in soil fertility (Zimov et al., 1995; Johnson, 2009); factors promoting increased methane emissions. There is good evidence that the near ubiquitous presence of black mats (a thin layer of organic material created under moist conditions such as elevated water tables, ponds, bogs and wet meadows) across North America dates to just after the megafaunal extinction (Haynes, 2008).

Today of course, megafauna have largely been replaced across the earth by livestock (Smith et al., 2016a, 2016b; Bar-On et al., 2018). Indeed, as the overall biomass of mammals on the planet has grown fourfold over the past few hundred years, the biomass of wild mammals has decreased by a factor of 6 (Bar-On et al., 2018). There

are now more than 1.5 billion cattle, 1.2 billion sheep, 1 billion goats, >950 million pigs on Earth (Robinson et al., 2014). In part because of these increases in livestock cultivation and industrialization, the methane concentration in the atmosphere has risen dramatically in recent centuries and is now 162% higher than pre-industrial levels (https://www.noaa.gov/news-release/increase-in-atmospheric-methane-set-another-record-during-2021). Globally, animal agriculture contributed about 30% of human-derived methane emissions in 2021 (https://www.ccacoalition.org/en/resources/global-methane-assessment-full-report). For example, of the 650 million metric tons of carbon equivalent of methane the United States released in 2021, >40% was derived solely from livestock emissions and manure management (https://www.epa.gov/ghgemissions/inventory-us-greenhouse-gas-emissions-and-sinks).

### *Redistribution of nutrients*

As we discussed earlier (see section 'Distance'), the fossil record for some of the largest Pleistocene megafauna suggests they had virtually continental-wide distributions (FAUNMAP Working Group, 1994; Lyons and Smith, 2013). This is not surprising because space use is positively related to body size (Brown, 1995). For example, even within their vastly restricted modern geographic distribution, elephant home range can reportedly exceed 8,500 to 10,000 $km^2$ (Lindeque and Lindeque, 1991; Ngene et al., 2017) and some individuals have been logged traveling more than 2,800 km annually (Mills et al., 2018). As animals travel through the landscape they also engage in activities such as foraging, digestion and ultimately, defecation. Because the passage rate of food through the body is positively associated with body size (Peters, 1983), elimination by megaherbivores occurs at some distance from where food was consumed (Doughty et al., 2016b; Veldhuis et al., 2018). Moreover, the digestion process liberates nutrients within vegetation that would otherwise be 'locked' within plant biomass for many decades (Owen-Smith, 1988; Johnson, 2009). Thus, the extensive lateral movement of megafauna can play a key role in the redistribution of nutrients such as carbon, sodium, phosphorus, and nitrogen and is particularly important in nutrient-poor systems or low-productivity environments (McNaughton et al., 1997; Doughty et al., 2013, 2016b, c; Schmitz et al., 2014; Veldhuis et al., 2018).

Megafauna can also transport nutrients across habitat types. For example, today hippos contribute to the lateral transfer and recycling of silicon by foraging on riverine grasslands, but defecating in rivers and ponds (Schoelynck et al., 2019). Conversely, by foraging on aquatic vegetation, but defecating on land, moose transfer aquatic-derived nitrogen to terrestrial ecosystems (Bump et al., 2009). The mass drownings associated with the annual migration of wildebeests result in the transfer of carbon, nitrogen and phosphorous from the Serengeti into the Mara River watershed (Subalusky et al., 2017). And, brown bears transport marine-derived nutrients to terrestrial environments in areas with large salmon runs (Hilderbrand et al., 1999). All of these processes significantly alter primary productivity and biodiversity in environments. Such enhancement of biogeochemical cycling is not replicated by small-bodied herbivores who eat less, have shorter passage rates, and much more restricted home ranges.

Several studies have suggested that the extinction of large-bodied mammals in the terminal Pleistocene fundamentally changed the pattern of nutrient deposition on landscapes, leading to declines in the fertility of terrestrial, aquatic and marine environments (Doughty et al., 2013, 2016b, c). For example, Doughty

et al. (2016c) explored the role of megafauna in the distribution of sodium on the landscape. Using an allometric-based modeling approach, they found significant differences in the spatial distribution of sodium related to the loss of megafauna. The lack of lateral transport by megafauna has resulted in modern ecosystems with higher sodium concentrations along coastal environments and reduced concentrations inland. Similarly, the fertility of oceans and terrestrial surfaces has likely decreased significantly since the late Pleistocene (Doughty et al., 2016b). Using a random walk mathematical formulation, Doughty et al. (2016b) computed that across the earth, the ability of animals to transport nutrients away from a point source has decreased to 6% of its former capacity. Overall, before the late Pleistocene extinction, the long-range movements of large-bodied mammals led to a more fertile planet.

Changes in behavior or foraging patterns can also alter the patchiness of nutrients on landscapes. Because of their large size, megaherbivores are mostly immune to predation and so not unduly influenced by the distribution of predators on the landscape. However, smaller-bodied herbivores do respond to the 'landscape of fear' (Brown et al., 1999; Laundré et al., 2001) and when predators are present they tend to curtail movement through areas perceived as higher risk (Le Roux et al., 2018). This leads to concentrations of dung, and hence nutrients, in areas where mesoherbivores aggregate, which tend to be more open and have greater visibility. Thus, in a world without megaherbivores, there is limited nutrient diffusion away from these fear-driven prey aggregations (Le Roux et al., 2018).

## Surviving mammals

The size-selective nature of the late Pleistocene extinction meant that energy flow within the community was fundamentally altered. An assumption underlying ecological theory is that energy does not go unused (Ernest and Brown, 2001), so this suggests that surviving species had a number of potential options – they could persist without change, alter their diet and/or morphology to better exploit newly available resources and/or become more abundant or widespread in the landscape. It is also possible that other taxonomic groups co-opted the energy that once flowed through megafauna, or that the 'excess energy' in the ecosystem allowed invasive species to establish. Here, we explore potential dietary, morphological and distributional changes in the surviving mammals.

### Dietary changes

When species are lost, ecological vacancies are created, which presents opportunities for survivors. In perhaps the most famous example, extinction of the non-avian dinosaurs ~65 Ma allowed mammals to expand into the ecological niches left vacant and ultimately led to their radiation and the many mammalian forms we see today (Alroy, 1998; Smith et al., 2010; Grossnickle et al., 2019). The loss of megafauna in the Americas at the terminal Pleistocene may also have provided many ecological opportunities for surviving species (Martin and Klein, 1984; Barnosky et al., 2004; Lyons et al., 2004, 2016a; Koch and Barnosky, 2006; Smith et al., 2016a, 2022; Tóth et al., 2019). In this section, we explore whether the extinction led to dietary shifts and/or expansion in surviving mammals. Did the survival of herbivore and carnivore species fill the roles left by the loss of their larger counterparts? We focus on inferences determined from geochemical proxies, namely stable isotope analysis (Box 1), which has rapidly become a common method to examine the present and past ecology of animal

---

**Box 1.** Stable isotope analysis.

Stable isotope analysis involves measuring the relative abundances of different *stable* isotopes in an organic sample, such as the ratio of $^{13}C{:}^{12}C$ or $^{15}N{:}^{14}N$. Note that these measurements are distinct from radiogenic dating methods which rely on the decay of *unstable* isotopes such as $^{14}C$. Stable isotope ratios are reported in delta values – $\delta$ – defined in units of parts per thousand, or per mil (‰) (Sharp, 2017). All samples and reference materials are calibrated against international standards to allow comparison among laboratories and analytical methods. The accepted standards for the two most commonly measured isotope systems in biological studies – $\delta^{13}C$ and $\delta^{15}N$ – are Vienna-Pee Dee Belemnite (VPDB) and atmospheric $N_2$, respectively.

The utility of a stable isotope methodology in biology and ecology relies on a few key observations. Firstly, carbon isotope values ($\delta^{13}C$) vary among different types of primary producers due to differences in photosynthetic mechanisms, biochemistry and growth rates (Fogel and Cifuentes, 1993). In terrestrial environments, plants using $C_4$ photosynthesis (e.g., grasses, corn, sugarcane and many monocots) have higher $\delta^{13}C$ values by 10–15‰ relative to plants with the $C_3$ photosynthetic pathway (e.g., trees, rice, most dicots; Ehleringer and Cerling, 2002). Consequently, $\delta^{13}C$ values in consumers provide a record of habitat use and of browse versus graze in extinct and extant animals (Ehleringer and Cerling, 2002). $\delta^{15}N$ values of consumers can be used to understand relative trophic positioning, as $^{14}N$ is preferentially excreted (Vanderklift and Ponsard, 2003), and so $\delta^{15}N$ values tend to increase with increasing consumer trophic level.

One important consideration is that both carbon and nitrogen values at the base of the food web are strongly influenced by climatic variables (Austin and Vitousek, 1998). Nonetheless, isotopic analysis allows researchers to characterize both the overall structure of local food webs as well as the flows of energy from basal production sources through to top consumers as long as these factors are considered (Ben-David and Flaherty, 2012).

In paleoecological or zooarchaeological studies, taphonomy must also be considered, as original tissue must remain in order to infer diet/ecology of ancient animals through stable isotope analysis. Clementz (2012) and Koch et al. (2017) present useful summaries of substrates that can be used for isotopic analysis of vertebrate paleoecology. Of these, collagen extracted from bone or dentine is among the most widely used, as this substrate contains carbon and nitrogen which can be measured for stable isotope ratios and used to assess protein diagenesis via evaluation of [C]:[N] atomic ratios (Ambrose, 1990).

---

populations (e.g., Newsome et al., 2007; Clementz, 2012; Layman et al., 2012; Koch et al., 2017).

As noted elsewhere, large-bodied herbivores were the most effected by the late Pleistocene extinction (Lyons et al., 2004; Smith et al., 2018). In North America, survivors included pronghorn (*Antilocapra americana*), several cervids (*Odocoileus virginianus, Odocoileus hemionus, Rangifer tarandus, Alces alces, Cervus elaphus*), bovids (such as *Ovis canadensis, Oreamnos americanus* and *Ovibos moschatus*) and a smaller species of bison (*Bison bison*). The next largest surviving herbivore, at just over 30 kg, was a peccary (Smith et al., 2004, 2019). With the loss of so many competitors, resource and habitat availability likely increased dramatically. As the majority of extinct taxa were grazers on $C_4$ vegetation (Smith et al., 2022), leaving this guild particularly depauperate, there was great opportunity for ecological expansion into a $C_4$ grazing niche by surviving species.

Despite this, little evidence exists for dietary shifts among the surviving large-bodied herbivores. For example, there was no change in the mean $\delta^{13}C$ value of bison (*B. bison*), deer (*Odocoileus* sp.) or antelope (*A. americana*) following the extinction in central Texas (Smith et al., 2022). Thus, somewhat paradoxically, surviving herbivore species apparently did not take advantage of the vacant ecological/resource space left by the extinct megafauna. However, while the extinction led to a disproportionate loss of grazers (Smith et al., 2016), multiple browsers (e.g., mastodon and flat-headed peccary) and mixed feeders (e.g., camels and gomphotheres) were

also lost, which likely reduced the competitive pressure on survivors across all herbivore foraging guilds. Additionally, C₄ grazers in central Texas likely fed predominantly on grasses (Smith et al., 2022), which are difficult to digest and require specialized physiology, such as foregut fermentation, hypsodonty dentition, increased gut retention times and overall larger gut capacity (Chivers, 1994; Stevens and Hume, 2004; Janis, 2008). This may have prevented or slowed surviving browser species from switching/expanding their diet. In contrast, grazers may be more readily able to switch to less recalcitrant browse (Bergmann et al., 2015; Craine, 2021; Pardi and DeSantis, 2021). Indeed, morphological studies suggest bison were more ecologically flexible in the late

Pleistocene than their modern congeners today (Rivals et al., 2007; Rivals and Semprebon, 2011; Kelly et al., 2021).

Among small herbivores, a group that emerged from the Pleistocene–Holocene transition largely unscathed, isotopic paleoecology studies demonstrate a high level of adaptive capacity and mixed responses to the extinction (Tomé et al., 2020, 2022; Smith et al., 2022). For example, in central Texas, several herbivorous rodents – the cotton rat (*Sigmodon hispidus*) and woodrat (*Neotoma* sp.) – responded quite differently to the Pleistocene–Holocene transition (Figure 5). The $\delta^{13}C$ values of *S. hispidus* tracked the relative proportion of local grazers, frugivores and granivores, as well as minimum temperature (Tomé et al., 2020).

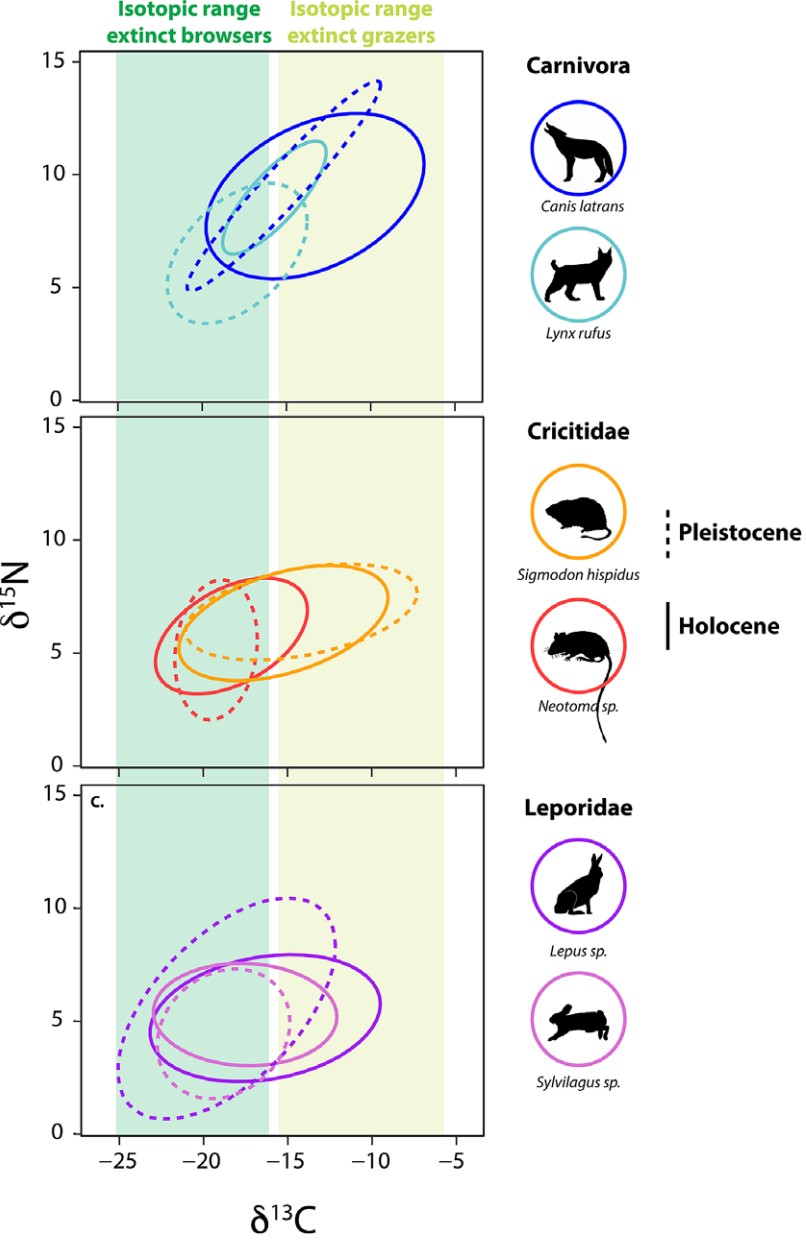

**Figure 5.** Bulk $\delta^{13}C$ and $\delta^{15}N$ values from bone collagen of mammalian survivors pre- and post- the terminal Pleistocene megafauna extinction. Data from Smith et al. (2022) and references therein. Ellipses represent the bivariate isotopic space occupied by each group (function stat_ellipseProgram R). Dashed lines show isotopic niche space of species prior to the extinction event, solid lines show isotopic niche post-extinction in the Holocene. Colored bars show the range of $\delta^{13}C$ values of extinct grazers and browsers. Animal silhouettes come from Phylopic; illustration credit to Nina Skinner, C. Camilo Julián-Caballero and Gabriela Palomo-Munoz.

However, isotopic values of *Neotoma* predominantly tracked climatic and vegetational changes (Tomé et al., 2022); there was also likely a species replacement from *N. floridana* to *Neotoma albigula/micropus* as Texas habitats shifted toward more warm and arid environmental conditions. Among lagomorphs, mean $\delta^{13}C$ values of both *Lepus* sp. and *Sylvilagus* sp. increased significantly following the extinction, a finding consistent with greater consumption of $C_4$ resources (Figure 5; Smith et al., 2022). In sum, trends demonstrate the complex responses by animals to the multiple effects of the megafauna extinction and highlight the varying ecological adaptations of both large and small mammals.

Differential dietary responses to the late Pleistocene extinction have also been observed in carnivore guilds, which likely relate to differences in ecological specialization. It appears that carnivores with broader diets had better outcomes (Galetti et al., 2018; Smith et al., 2022), which may explain the large dietary breadth found among most carnivores today. Among the Canidae, neither coyotes (*Canis latrans*), or wolves (*Canis lupus*) in central Texas exhibited changes in their isotopic niche following the extinction (Smith et al., 2022). Similar results were found for gray wolves in the Yukon Territory of Canada (Landry et al., 2021), although DeSantis et al. (2019) reported changes in coyote foraging behavior following the megafauna extinction in coastal southern California. In the central Texas community, coyotes and wolves consistently displayed wide ranges in measured isotope values, suggesting intrinsically high degrees of ecological and dietary plasticity in both the Pleistocene and Holocene (Smith et al., 2022). In contrast, fossils of the extinct dire wolf, *Aenocyon* (*Canis*) *dirus*, from the same region exhibited a narrower isotopic niche space (Smith et al., 2022). This dietary (isotopic) variability in coyotes and *C. lupus* may have buffered these species from extinction risk, but also made it difficult to detect any directional changes in diet. Similarly, no ecological shifts were noted in mesocarnivores (e.g., Mephitidae and Procyonidae) or bears (Ursidae) following the extinction (Smith et al., 2022).

In contrast, the megafauna extinction led to highly significant isotopic changes within central Texas Felidae (Smith et al., 2022). In the Pleistocene, the two sabertooth cats (*Homotherium serum* and *Smilodon fatalis*) were apex carnivores. Both sabertooth cats had high $\delta^{13}C$ and $\delta^{15}N$ values and fairly narrow isotopic ranges indicative of specialization on large grazers (Smith et al., 2022). Enamel $\delta^{13}C$ values from *H. serum* and *S. fatalis* found in La Brea, California also suggest a high degree of dietary specialization (DeSantis et al., 2019, 2021). Following the extinction of these large cats in Texas, the surviving felids shifted in bivariate stable isotope space (Smith et al., 2022). Lynx (*Lynx rufus*) and jaguar (*Panthera onca*) exhibited higher $\delta^{13}C$ values following the extinction with *L. rufus* also having higher $\delta^{15}N$ values in the Holocene (Figure 5). *P. onca* became the new apex predator in the system, taking over what was left of the ecological niche space once occupied by the extinct cats. The mountain lion (*Puma concolor*) also appears in the community by the early Holocene. These findings support the idea of 'mesopredator release' (i.e., the expansion in diet, range and/or abundance of a smaller predator following the reduction or removal of a larger congener; Soulé et al., 1988; Polis et al., 1989) within the Felidae. However, the ecological role of the surviving predators was likely vastly different from the megacarnivores they 'replaced'. Indeed, because these new apex predators were so much smaller, they would have fed on smaller and different prey (Carbone et al., 1999; Hayward, 2005, 2006, 2016).

The dietary responses of surviving species to the ecological vacancies left by the extinction of megafauna were clearly multifaceted and varied widely depending on trophic level and dietary guild. Upon reflection, this result is unsurprising, as food webs are complex, highly localized systems held in a delicate balance by direct and indirect interactions among species (e.g., Chesson and Kuang, 2008). It is also important to note that the patterns we describe here are based on a handful of studies and, in some cases, a handful of data points. Broad trophic guild categorizations like carnivore, grazer and browsers, often do not adequately capture the dietary complexity of many taxa, particularly when sample sizes are small. Thus, we encourage our colleagues, or any aspiring historical or paleoecologist, to employ a geochemical (i.e., stable isotope analysis; Box 1) perspective to ask similar questions in other localities in North and South America (see Smiley et al., 2016; Terry, 2018; Taylor, 2019). Interesting macroecological patterns in species responses may emerge with additional datasets, as will a more comprehensive understanding of the full range of ecological plasticity in our extant mammalian fauna.

### Morphological changes

It is generally recognized that body mass is the most important trait of an animal (Thompson, 1942; Bonner, 2006; Smith and Lyons, 2011, 2013; Smith et al., 2016d; Smith, 2021). The body size of a mammal is not only highly heritable (Smith et al., 2004), but also strongly influences all biological rates and processes, including the essential activities of metabolism, reproduction and growth (Peters, 1983; Calder, 1984; Schmidt-Nielsen, 1984). Because of the critical role body size plays in governing the physiology, life history, behavior and ecology of animals, it mediates how they interact with other animals and their environment (Smith, 2021).

Morphological shifts are a common response to changing environmental conditions across spatial gradients and over evolutionary time (James, 1970; Smith et al., 1995; Smith and Betancourt, 1998, 2003, 2006; Millien et al., 2006; Gardner, 2011; Secord et al., 2012; Balk et al., 2019) and are also influenced by ecological factors such as competition (e.g., Hutchinson and MacArthur, 1959; Peters, 1983). Certainly, body size plays an important role today in structuring mammal communities in Africa (De Iongh et al., 2011). As new niches opened up following the megafauna extinction, one response may have been for the surviving lineages of smaller-bodied animals to become larger so as to better exploit – either physiologically or ecologically – these newly available resources. Thus, here we explore shifts in body size among survivors of the late Pleistocene megafauna extinction. Body size is fairly easy to characterize with fossils because virtually all cranial and postcranial elements scale with mass in mammals (e.g., Damuth and MacFadden, 1990; Smith, 2021; Smith et al., 2022).

While considerable work has focused on the morphological responses of animals to the climate changes of the late Quaternary (e.g., Smith et al., 1995; Smith and Betancourt, 1998, 2003, 2006; Barnosky et al., 2003; MacDonald et al., 2008; Gardner, 2011; Hoffman and Sgrò, 2011), few studies to date have explicitly examined how biodiversity loss at the terminal Pleistocene may have driven body size changes (but see Tomé et al., 2020a, 2022; Smith et al., 2022). At our Hall's Cave site in central Texas, we find no consistent pattern in morphological responses to the biodiversity loss at the late Pleistocene. Indeed, the shifts in morphology from the Pleistocene to Holocene were quite species-specific, with some animals (i.e., deer and jackrabbits) becoming significantly larger, and others significantly smaller (i.e., bison and cottontails), and still others exhibiting no change in body size at all (i.e., pronghorn, raccoons and coyotes; Smith et al., 2022). These results suggest that most species were responding to other factors,

and that if they responded to biodiversity loss, it was in other ways – perhaps through changes in abundance or distribution (see section 'Changes in distribution and abundance').

Interesting exceptions include several small mammal taxa that did exhibit changes in morphology associated with the extinction. For example, changes in community structure (i.e., alpha and beta diversity, proportion of each tropic guild) were associated with changes in body size and diet in the hispid cotton rat, *S. hispidus.* Prior to the extinction when the community had many grazers, insectivores and browsers, cotton rats were smaller. As the community composition shifted toward a greater proportion of browsers and omnivores, the body size of cotton rats increased. The changes in morphology seen over this time may reflect shifts in resource availability (Tomé et al., 2020a). But body size responses were not consistent across small-bodied taxa. While *Sigmodon* was quite sensitive to community turnover and associated vegetation change, the morphology of other small-bodied rodents such as *Neotoma* were strongly correlated to climate change at the site and not biodiversity (Tomé et al., 2020a, 2022). As yet, we have not characterized responses by the entire small-bodied mammal guild, so it is unclear whether patterns may yet prove to be consistent along trophic affiliation, body size or other ecological gradients. Such individualistic body size responses to biodiversity loss demonstrate the complex nature of communities and ecological interactions within them. Future work should investigate whether there is a regularity to how taxa respond morphologically to biodiversity loss across a spectrum of body sizes (i.e., shrews to deer), and habitat and resource preferences (i.e., deserts, grasslands and forest).

## Changes in distribution and abundance

Range shifts were common during the late Pleistocene and Holocene (Grayson, 1987, 2000; Graham et al., 1996; Lyons, 2003). Because the geographic distribution is a dynamic reflection of a taxon's realized ecological niche, which is constrained by physiological tolerances to the abiotic environment as well as interactions with other species (Grinnell, 1917; Hutchinson, 1957; Loehle and LeBlanc, 1996), it is particularly sensitive to changes in abiotic and biotic factors (Brown et al., 1996). The late Pleistocene megafauna extinctions were accompanied by significant climate change meaning that both factors may have influenced range shifts (Grayson, 1987, 2000; Graham et al., 1996; Lyons, 2003).

Quantification of mammal species range shifts across the Pleistocene–Holocene transition found that the centroid of species ranges shifted in different directions, and to different extents; moreover, the magnitude of the change in range size differed among species (Lyons, 2003). These differences highlight the individual responses of species to ecological change, as evidenced in both body size and diet shifts (see sections 'Dietary changes' and 'Morphological changes'). We can see this clearly at Hall's Cave. Currently, this site lies outside of the modern geographic range of margay (*Leopardus wiedii*), white-throated woodrat (*N. albigula*), Southern bog lemming (*Synaptomys cooperi*), meadow jumping mouse (*Zapus hudsonius*), the common shrew (*Sorex cinereus*), prairie shrew (*Sorex haydeni*), Southeastern shrew (*Sorex longirostris*) and the Southern short-tailed shrew (*Blarina carolinensis*), yet all these species co-occurred during the Pleistocene (Smith et al., 2016c). Examination of the fossil record of two of these species (*S. cooperi* and *B. carolinensis*; Figure 6), shows that while neither are found together today (and indeed, they have quite distinct distributions), they were both present at Hall's Cave in the Pleistocene. Over the Holocene, *S. cooperi* shifted the southern boundary of its geographic range further north, while the western boundary of *B. carolinensis* shifted east.

To what extent were such shifts in the geographic range of mammals driven by biodiversity loss? Niche modeling of late Pleistocene and early Holocene distributions indicates that climate tracking drove some, but not all, of the observed changes, suggesting a role for biotic interactions (Martínez-Meyer et al., 2004; Veloz et al., 2012; Maguire et al., 2015; Pardi and Smith, 2016). At the end of the Pleistocene, the largest surviving herbivores lived in communities with fewer competitors and fewer large predators; thus, changing biotic interactions likely impacted range shifts. Herbivores respond to resource availability, but they also compete with one another and increase vigilance and avoid areas where they are susceptible to predation (Brown et al., 1999). The presence of large predators influences herbivore density and their use of the landscape, and thus can affect herbivory pressure and plant communities (Ripple et al., 2001; Ripple and Beschta, 2003). A macroecological approach assessing carnivore body size and maximum prey size (Van Valkenburgh et al., 2016) demonstrated that large Pleistocene carnivores could have hunted megaherbivores and effectively limited their populations. This enhanced landscape of fear would have been felt throughout ecosystems, and top-down forces likely had strong impacts on the abundance and distributions of some species (Ripple and Van Valkenburgh, 2010).

The megafauna extinction brought with it the loss of interactions with large competitors. Carnivores experience strong intraguild interactions; they not only compete but also engage in frequent intraguild killing where the larger species is typically dominant (Polis, 1981; Polis et al., 1989). Competition between carnivores and the ability to defend territory are strong forces dictating geographic distributions, to the extent that smaller carnivores will avoid the territories of larger competitors (Polis et al., 1989; Fedriani et al., 2000; Aunapuu et al., 2010). As a result, the extinction of the very largest carnivores at the end of the Pleistocene likely changed these interactions (see section 'Dietary changes'), and impacted the geographic distribution of surviving carnivore species. For example, three large canid morphotypes existed at the end of the Pleistocene (i.e., dire wolves, gray wolves and Beringian wolves) and partitioned the landscape, at least in part, through competitive exclusion (Dundas, 1999; Meachen et al., 2016). Ecological niche modeling has demonstrated that the geographic ranges of surviving Canidae did not track climate change following the extinction (Pardi and Smith, 2016). Instead, changing interactions, such as release from intraguild competition with congeners, was important. Body size played a role in these changes as the ranges of the largest survivors, wolves and coyotes, deviated the most from climate predictions; in contrast, the smallest survivors, foxes, more closely tracked Holocene climate shifts (Pardi and Smith, 2016).

Another probable response to the biodiversity loss in the late Pleistocene was numerical; taxa may have increased in abundance to exploit newly available resources. The absence of millions of large-bodied mammals on the landscape left many potentially vacant ecological niches. However, abundance is notoriously difficult to characterize in the fossil record because of taphonomic biases and thus little work has explored this. Interestingly, recent modeling efforts of bison – the largest bovid that survived the extinction in North America – suggests intriguing changes in the abundance and distribution of bison herds over the late Pleistocene and Holocene (Martin et al., 2022; Wendt et al., 2022). Prior to the

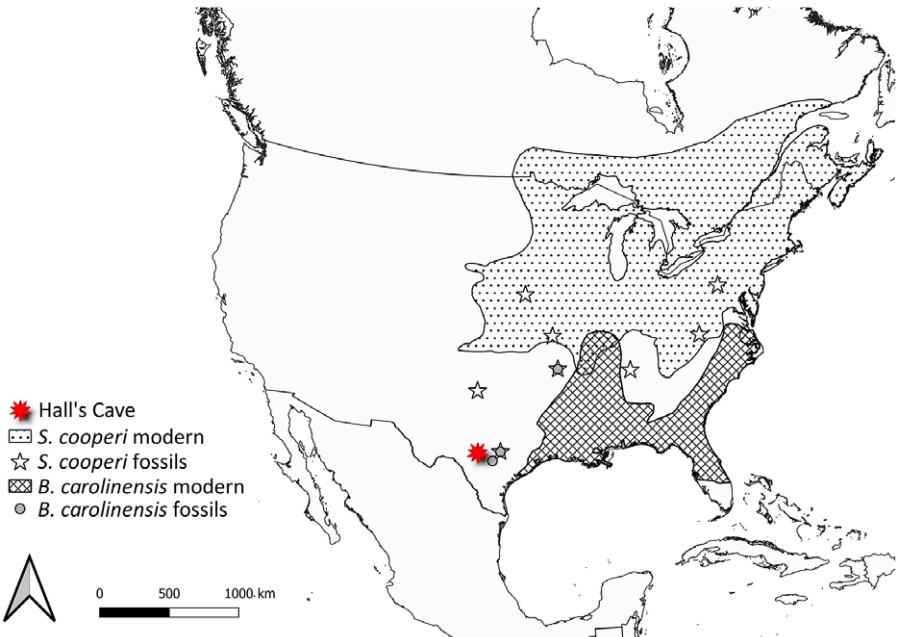

**Figure 6.** Fossil localities and modern distributions of *Synaptomys cooperi* and *Blarina carolinensis*; examples of species found outside of their modern geographic distribution at Hall's Cave. These species occurred at Hall's Cave in the past and have changed their respective geographic ranges individualistically. Late Quaternary fossil occurrences come from the Neotoma database (Williams et al., 2018) with modern distributions superimposed (Marsh et al., 2022).

megafauna extinction, animals were widely scattered across North America into regional clusters. This changed by the late Holocene when the bison were spatially contiguous through much of central and western North America (Wendt et al., 2022). While the authors attribute much of the shifts in both distribution and abundance of bison herds to tracking of favorable environmental conditions, they do note a spike in abundance at the terminal Pleistocene and early Holocene (Wendt et al., 2022). These increases in distribution and abundance in the early Holocene may have represented expansion due to the loss of competitors (Wendt et al., 2022).

Species adaptations and range shifts are highly individualistic responses to biodiversity loss and environmental change; taking both possibilities into consideration demonstrates the complex nature of communities and ecological interactions within them. To understand how animal morphology may respond to extinction, future work needs to examine responses across a spectrum of body sizes (i.e., shrews to deer), and habitat and resource preferences (i.e., deserts, grasslands and forest). Further advances in occupancy modeling should allow the characterization of abundance shifts in species.

## Changes in communities and ecosystems

Modern studies demonstrate that the extirpation of the largest animals in a community profoundly changes the ecological interactions and behavior of surviving animals and the energy flow through the ecosystem (Estes et al., 2011); thus, the late Pleistocene extinctions likely led to similar perturbations. These might include shifts in biotic interactions (i.e., competition and predation), food webs and aspects of community structure such as co-occurrence patterns and functional diversity. We discuss changes in ecological interactions among taxa and the shifts in the structure of ecological communities below. Where possible, we use Hall's Cave as a case study to contextualize patterns found at other sites and scales.

### Ecological interactions within communities

The loss of megafauna led to profound changes in ecological interactions within communities. Not only were some biotic interactions entirely lost when both species were extirpated, but many interactions also involving surviving species and (now) extinct ones were also lost. This led to a cascade of effects that percolated throughout the community and included changes to the connectivity and stability of food webs (Nenzen et al., 2014; Pires et al., 2015; Fricke et al., 2022). We discuss these below.

### Species interaction networks and food webs

The loss of megafauna from ecosystems had numerous and diverse effects on species interactions. The most obvious, of course, was the loss of biotic interactions between megafauna and other mammals, which resulted in food webs with fewer connections between taxa (Nenzen et al., 2014; Fricke et al., 2022). However, the effects were exacerbated because of the size-selectivity of the extinction. This is because the reduction of connectivity in food webs is not directly proportional to biodiversity loss; large-bodied mammals form more links per species than do smaller-bodied mammals (Sinclair et al., 2003; Owen-Smith and Mills, 2008; Pires et al., 2015; Fricke et al., 2022). Thus, the disproportionate extirpation of large-bodied mammals in the late Pleistocene meant food webs lost many more links than expected (Fricke et al., 2022). Current biodiversity loss is continuing this trend; the extinction of remaining megafauna will have an overwhelming effect on Earth's already depauperate food webs (Fricke et al., 2022).

The reorganization of food webs began after human arrival on a continent and intensified after the start of European colonization and the industrial revolution (Fricke et al., 2022). For example, despite being robust to secondary extinctions for 850,000 years, extinctions significantly altered mammal food webs in Iberia in the Holocene (Nenzen et al., 2014). The loss of large-bodied mammals led to less robust food webs, which were less resilient to future

extinction (Nenzen et al., 2014; Pires et al., 2015). Moreover, the type of mammal lost matters; large predators tend to have broad dietary niches and interact with a large number of other species. Thus, they have higher per capita interaction strengths than large herbivores or smaller predators. As a result, the addition of a large novel predator like humans had a greater effect in communities with a diverse predator guild such as the late Pleistocene of North America (Pires et al., 2015). South American communities had fewer large predators relative to their large herbivore guild (Fariña, 1996; Fariña et al., 2013). Not surprisingly, then, the greater proportion of carnivore extinctions in North America led to a greater decrease in stability than in South America (Pires et al., 2015). African communities underwent a reduction in large carnivore diversity earlier in the Pleistocene, arguably due to competition with early *Homo* species (Lewis and Werdelin, 2007; Werdelin, 2012; Werdelin and Lewis, 2013) leaving modern African communities with a relatively depauperate large predator guild. Hence, present day African communities are more robust to the extinction of a similar magnitude (Pires et al., 2015).

Another reason the terminal Pleistocene extinction left such lasting effects on modern mammalian food webs is because of the unique functional roles of the species that went extinct (see section 'Functional changes within communities' on functional diversity). Because large-bodied herbivores are often ecosystem engineers (Owen-Smith, 1988), their loss has cascading effects on surviving species (e.g., Estes et al., 2011; Ripple et al., 2015). For example, as discussed in Section 'Dietary changes', an in-depth analysis of the mammal community at Hall's Cave using stable isotopes (Box 1) found that, with the exception of the felid guild, the dietary niches vacated by extinct large-bodied species were not refilled (Smith et al., 2022). The isotopic niche occupied by mammoth, mastodon, horses, camels, and extinct pronghorn and bison remained empty; surviving canids did not colonize the niche previously occupied by the dire wolf (*Aenocyon* (*Canis*) *dirus*). However, a trophic cascade was likely caused by the loss of the apex felid predators (i.e., *Smilodon fatalis, Homotherium serum, Pathera leo atrox*; Smith et al., 2022). This led to the jaguar (*P. onca*), a former mesopredator, moving into the apex predator role and may have had impacts on species interactions at every level of the community (e.g., Estes et al., 1998, 2011, 2016).

Although the temporal resolution is lacking to track these hypothesized trophic cascades across the entire Holocene, the mammal community at Hall's Cave, Texas likely experienced a profound and lasting loss of ecological complexity (Smith et al., 2022). Simply tracking the loss or shifts in dietary niches does not yield a complete picture of how the extinction altered the community. Mammals with seemingly similar dietary niches based on stable isotope analyses or traditional ecological categories (e.g., Elton traits; Wilman et al., 2014) can coexist if they further partition their niche based on body size (e.g., De Iongh et al., 2011). Prior to the extinction, Hall's Cave functioned in a similar way. In particular, browsers overlapped greatly in isotopic niche space, but occupied a diverse set of body sizes (Smith et al., 2022). By combining information from both stable isotopes and body size, Smith et al. (2022) found that not only was there a reduction in dietary niche space, but also the variation in body sizes within niches was also reduced. If this trend was typical for mammal communities before and after the megafaunal extinction, and it likely was, it may explain much of the loss of biotic interactions (Tóth et al., 2019) and changes in species associations (Lyons et al., 2016b) found at other spatial scales (see section 'Species co-occurrence' for more information).

### Species co-occurrence

The loss of large-bodied mammals at the late Pleistocene profoundly altered the behavior of surviving mammals, and especially their ecological interactions with each other (Lyons et al., 2016b; Smith et al., 2016c; Tóth et al., 2019). The tendency for a species' presence to be correlated with the presence or absence of another species (i.e., form a 'pair') is known as 'co-occurrence' (Gotelli and Ulrich, 2010). For hundreds of millions of years, if two species formed a significant association, they tended to aggregate or co-occur (Lyons et al., 2016b). However, there was a fundamental reversal of this pattern in the mid-Holocene and today most significant species pairs are segregations (Lyons et al., 2016b). That this switch was likely tied to the extinction of megafauna was demonstrated by Tóth et al. (2019). By decomposing the association strengths of species pairs into abiotic and biotic components, they were able to assess the relative influence of these components on species co-occurrence patterns over the late Quaternary. They found that while the influence of climate was constant, the influence of biotic interactions declined significantly following the megafauna extinction (Tóth et al., 2019). Analyses at local scales have found similar patterns, which provide additional context for understanding these continental scale changes. For example, the co-occurrence structure of mammals within the Hall's Cave community in Texas also changed with the extinction (Smith et al., 2016c). Pre-extinction significant aggregations and segregations were common, although there were some differences among trophic guilds. In particular, extinct carnivores were much more likely to form aggregations than segregations, owing to tightly linked predator–prey pairs (Smith et al., 2016c). Co-occurrence patterns changed considerably after the extinction, with few significant pairs found among carnivores. This suggested profound changes in how ecological processes influenced the structure of the mammal community.

The loss of interactions within food webs (Nenzen et al., 2014; Pires et al., 2015; Fricke et al., 2022), ecological complexity (Smith et al., 2022) and biotic interactions (Tóth et al., 2019) over the late Quaternary has consequences for macroecological patterns at larger scales. For example, the relationship between trophic structure and body size in mammals forms a U shape when ordered along an axis of increasing protein in the diet (Figure 7). Indeed, the average body size of herbivores and carnivores is much larger than that of omnivores and invertivores (Hiiemae, 2000; Price and Hopkins, 2015; Pineda-Munoz et al., 2016; Cooke et al., 2022). This relationship is consistently found across vertebrate taxa including mammals, birds, and fish, and began in mammals at least as early as 66 Ma (Cooke et al., 2022). Mammals in different biomes also consistently show a U-shaped relationship between diet and body size suggesting this is a fundamental way for mammals to divide up energy within an ecosystem (Cooke et al., 2022). However, as a result of the megafaunal extinction, this pattern is beginning to flatten. Because the largest herbivores and carnivores were disproportionately lost, the average body size of these trophic guilds has decreased. Future extinctions are likely to exacerbate this process (Cooke et al., 2022).

### Changes in community structure

All of the responses documented in previous sections (e.g., changes in species movement patterns, foraging, nutrient cycling, species distributions and ecological interactions) have resulted in changes to the structure and function of ecological communities. In addition to being depauperate, mammal communities in the Holocene show

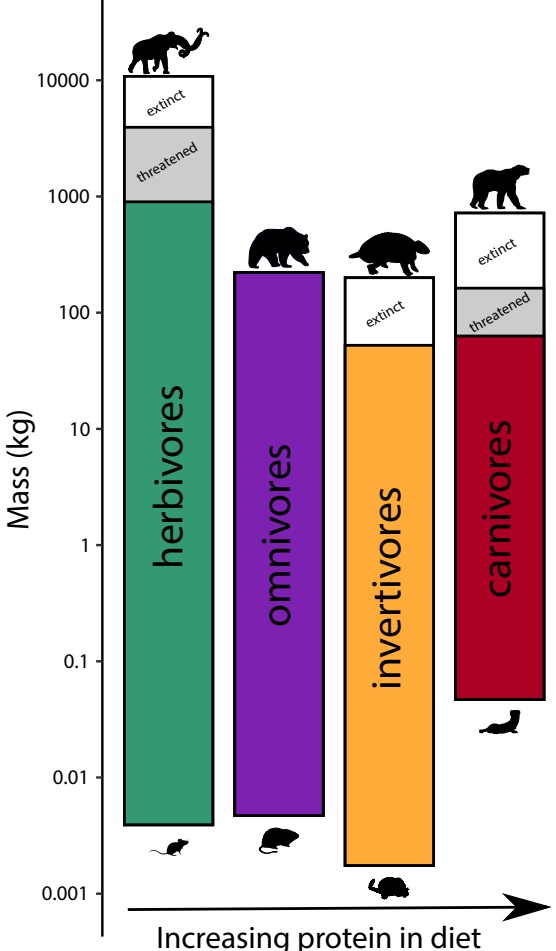

**Figure 7.** Relationship between body size and trophic guild for late Pleistocene Terrestrial mammals. Colored and gray boxes indicate the minimum and maximum sizes of extant mammals in each trophic guild. White boxes indicate maximum size when including extinct late Pleistocene species. The gray boxes indicate the decrease in body size expected in each trophic guild if all threatened and endangered species on the IUCN Red List go extinct. Data are from Smith et al., 2003. The largest and smallest species for each trophic guild are represented by silhouettes (PhyloPic.org). Creator credits: (top, left to right) U.S. National Park Service, xgirouxb, Xvazquez, Tracy A. Heath; (bottom, left to right) Daniel Jaron, Natasha Vitek, Becky Barnes, Ferran Sayol. Figure modified from Cooke et al., 2022.

changes in many of the standard metrics used to quantify community structure (Graham et al., 1996; Lyons and Smith, 2013; Lyons et al., 2019; Smith et al., 2019, 2022; Fraser et al., 2022; Hedberg et al., 2022). These include a profound increase in the similarity of communities across space (Graham et al., 1996; Fraser et al., 2022) and a loss of functional diversity (Davis, 2017; Hatfield et al., 2022; Hedberg et al., 2022).

*Community homogenization*
The severe size-selective biodiversity loss at the late Pleistocene led to increased homogenization of mammalian fauna and biogeographic provinces in North America over the Holocene (Graham et al., 1996). They speculated that this homogenization was due to climate change associated with deglaciation and the resulting changes in the environment (Graham et al., 1996). Fraser et al. (2022) tested this hypothesis by examining beta diversity of mammalian communities across the last 35,000 years in 5,000-year increments. Like Graham et al.

(1996), they found a significant decrease in beta diversity (i.e., species turnover) immediately post-extinction in North America (Fraser et al., 2022). However, they did not find an association between climate change and community homogenization. Ecological communities had retained their distinctiveness over the previous 20,000 years despite climate changes associated with deglaciation (Fraser et al., 2022). In addition, the magnitude of the homogenization was especially pronounced when considering only mammals larger than 1 kg, suggesting that the result was not due to reduced preservation potential of smaller-bodied species (Behrensmeyer et al., 2000; Miller et al., 2014). Thus, they concluded the underlying mechanism was biodiversity loss due to the extinction and the expansion of geographic ranges by survivors (Lyons, 2003; Tóth et al., 2019; Fraser et al., 2022). The increase in geographic range size was larger than expected based simply on the newly available land area due to the retreat of glaciation (Fraser et al., 2022). However, the infilling of species ranges did not increase; occupancy of species' geographic ranges was highly patchy (Tóth et al., 2019; Fraser et al., 2022). A second wave of homogenization began in the last 2,000 years coincident with the intensification of agriculture in North America at this time (Fraser et al., 2022). Although early agriculture did not produce monocultures, or alter the landscape to the extent that modern farming does (Smith, 1994; Price, 2009), early anthropogenic impacts including increasing human population size, landscape alteration and increasing land area under cultivation (Marlon et al., 2013) likely contributed to biotic homogenization in the late Holocene.

*Functional changes within communities*
Animals fill roles within ecosystems that vary in importance to the community. These are determined by the combination of their specific ecological or morphological traits (i.e., body size, diet, period of activity, social behavior; Box 2) and how these traits interact with their local environment (Owen-Smith, 1988; Geremia et al., 2019; Cox et al., 2021). Quantifying the range and distribution of functional traits within a community can provide a link between community composition and ecosystem function (Mouillot et al., 2011). In this section, we discuss changes in functional diversity across the terminal Pleistocene megafaunal extinction through the modern and explore how these are manifest at different spatial scales. We describe how shifts in functional composition may be

---

**Box 2.** What is functional diversity?

Functional diversity is typically quantified in multi-dimensional space constructed from relevant ecological trait axes. Taxa are plotted based on their trait values and various individual and community-level metrics calculated based on the position and distribution of the entire assemblage in functional space. Relevant metrics are listed below.

*Community metrics*
$FV_{OL}$: Functional volume, calculated using kernel density hypervolume estimation around all taxa, measures overall functional richness of a community (Blonder et al., 2018; Mammola and Cardoso, 2020).
$FR_{IC}$: Functional richness, calculated by measuring the volume of the minimum convex hull enclosing all taxa, measures maximum functional range of a community (Villéger et al., 2008).
$FD_{IS}$: Functional dispersion, calculated as the average distance of all taxa to the centroid, measures functional spread of a community (Laliberte and Legendre, 2010).
$FD_{IV}$: Functional divergence, calculated as the evenness in distances of all taxa to the centroid, measures functional symmetry of a community (Villéger et al., 2008).
*Individual metrics*
Functional distinctiveness: Mean distance to all other taxa, measures rarity of trait values compared to community as a whole (Violle et al., 2017).
Functional uniqueness: Distance to nearest taxon, measures similarity to nearest functional neighbor (Violle et al., 2017).

linked to demonstrated alterations in ecosystem function, including habitat structure, plant diversity, biogeochemical cycling and other topics of this review. Finally, we discuss the potential role of modern exotic and domesticated large mammals in restoring late Pleistocene functional composition and the degree to which they create novel communities.

During periods of biodiversity decline such as the terminal Pleistocene, changes in functional diversity can be decoupled from changes in species richness (Petchey and Gaston, 2002; Boyer and Jetz, 2014). Unlike taxonomic affiliation, which inherently assumes equal contribution to diversity, the functional contribution of an animal varies depending on the redundancy or distinctiveness of their ecological traits relative to the entire community (Mouillot et al., 2013a; Violle et al., 2017). This means that the consequences of extinction can be magnified – or ameliorated – by the particular subset of species that go extinct. Recent work exploring the ecological impacts of late Pleistocene biodiversity loss on communities has used a functional trait-based approach, allowing concrete ecological comparisons between communities across broad temporal and/or spatial scales (Dineen et al., 2014; Foster and Twitchett, 2014; Davis, 2017; Pimiento et al., 2017; Lundgren et al., 2020; Hedberg et al., 2022).

The late Pleistocene extinctions resulted in a severe decline in functional diversity at local scales (Hedberg et al., 2022). For example, at Hall's Cave, the loss of megafauna led to a significant reduction in both functional richness and volume within the community (Box 2; Figure 8A, B). Notably, the reduction in these metrics exceeded that expected if species loss were random, suggesting the large mammals extirpated contributed distinct ecological function (Hedberg et al., 2022). Indeed, an entire region of functional space was truncated (Figure 8E), leading to significant declines in functional dispersion and divergence (i.e., the spread and symmetry of taxa around the centroid of functional space, respectively) in the Holocene (Figure 8C, D; Hedberg et al., 2022).

In contrast, changes in functional diversity of the entire North American megafaunal assemblage at the continental scale were less acute across this same interval. While both functional richness and dispersion declined following the Terminal Pleistocene megafaunal extinction, the loss was within the expected range of random species decline (Davis, 2017). However, this analysis was limited to species greater than ~4 kg and declines in functional diversity became increasingly distinct from random expectation with lower minimum mass thresholds, supporting the notion that large mammals are indeed more functionally distinct compared to their smaller-bodied counterparts (Davis, 2017). Furthermore, large mammals make up a greater proportion of local versus regional or continental-scale biotas due to greater spatial turnover in small and medium-sized mammals (Brown and Nicoletto, 1991). Thus, the ecological impact of megafaunal species loss may be more evident at smaller, more local spatial scales.

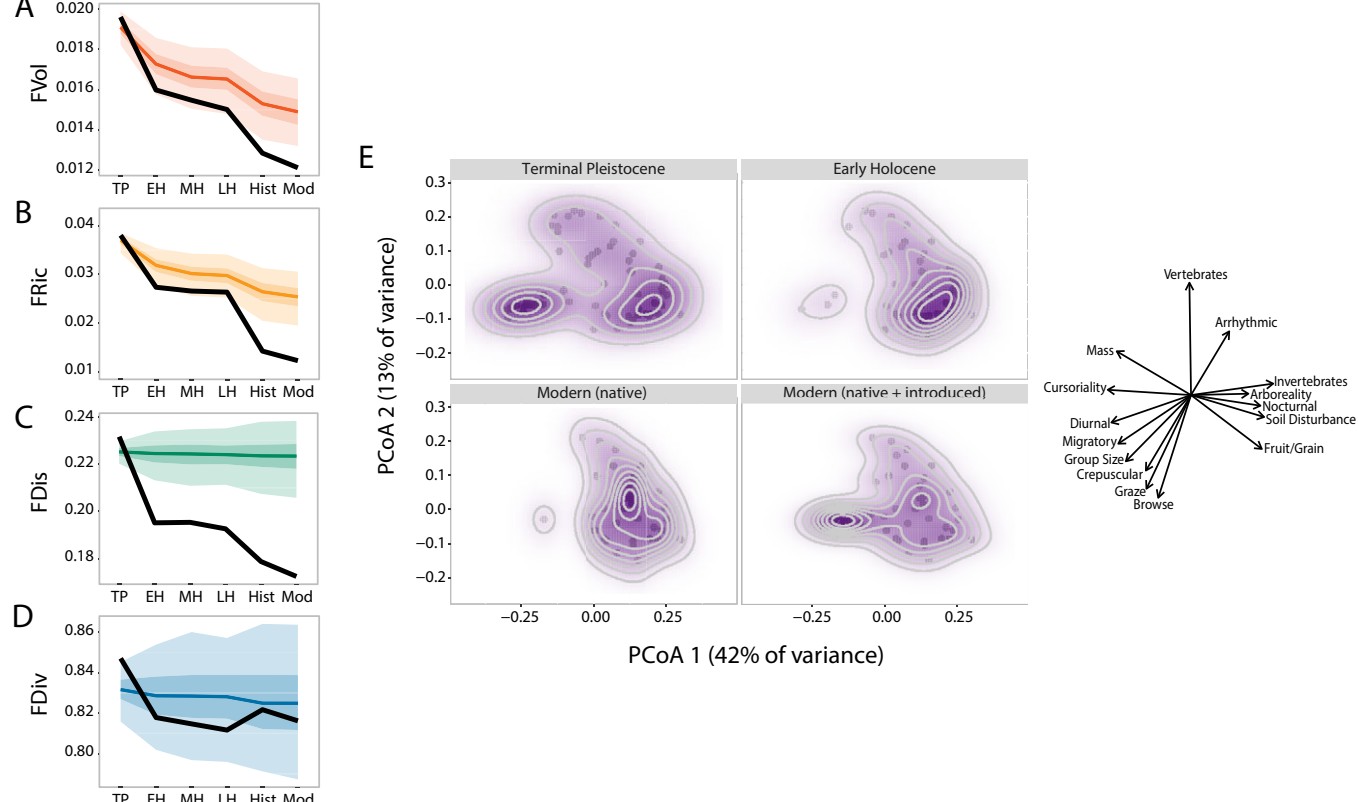

**Figure 8.** Key functional diversity metrics at Hall's Cave over time. (A) Functional volume (FVol). (B) Functional richness (FRic). (C) Functional dispersion (FDis). (D) Functional divergence (FDiv). Black lines represent empirical values; colored lines represent median values from null models simulating random species loss. Shading around colored lines indicates the 50% confidence interval (darker) and 95% confidence interval (lighter) of null model values. Terms: TP, terminal Pleistocene; EH, early Holocene; MH, middle Holocene; Hist, historical; Mod, modern. (E) Functional trait space of select temporal communities at Hall's Cave visualized on principal coordinates analysis (PCoA) axes 1 and 2 (55% of variance). Arrows on the right indicate how traits load onto each axis, with PCoA 1 primarily reflecting mass ($r = -.921$), invertebrate consumption ($r = .996$) and migration habit ($r = -.901$) and PCoA 2 primarily reflecting vertebrate consumption ($r = .999$) and browse ($r = -.914$) or graze ($r = -.833$) consumption. Each point represents a species, and fill color represents their density in functional space.

Perhaps more important than quantifying changes in aggregate functional diversity is understanding how the loss of megafauna shifted the distribution of ecological traits in surviving communities. For example, in addition to a significant truncation of the body size distribution (Smith et al., 2016c), the Holocene community at Hall's Cave shifted toward lower diurnal activity and lower prevalence of browse and graze consumption. Invertebrate and fruit and seed consumption became significantly more prevalent, as did fossorial and arboreal life modes (Hedberg et al., 2022). These changes in functional composition potentially signified a fundamental shift in energy flow through the ecosystem that could underpin changes in vegetation structure and composition (Asner et al., 2009; Bakker et al., 2016a). Indeed, the cave pollen record over this interval reflects an increase in tree abundance and a change from more open to closed habitats (Cordova and Johnson, 2019). Though climate was also in transition over the late Pleistocene, it is likely the loss of large-bodied herbivores that strongly contributed to this ecosystem shift. Non-random changes in trait-distributions were also observed at the continental scale, with the surviving biota similarly characterized by increased levels of fossoriality and arboreality and declines in graze and browse diets (Davis, 2017). Thus, the changes in local functional composition in central Texas were likely not unique, potentially leading to broader landscape-scale changes in ecological function. Indeed, documented cases of ecosystem transitions and changes in fire regimes and biogeochemical cycling following megafauna decline are detailed in Sections 'Foraging' and 'Digestion'. Further work exploring other communities is warranted.

Functional redundancy – a specific trait or suite of traits shared by multiple members of a community – can provide ecological resilience as a buffer against loss of function following extinction or other disturbances (Fonseca and Ganade, 2001; Mouillot et al., 2013a; Oliver et al., 2015, Biggs et al., 2020, Pimiento et al. 2020b). Redundancy is typically split into two metrics: functional distinctiveness and functional uniqueness, which refer to the mean distance to all other taxa and the distance to the nearest taxon in functional space, respectively (Violle et al., 2017; Box 2). As implied by the severe loss of functional richness and dispersion at Hall's Cave, Pleistocene megafauna were significantly more functionally distinct than surviving taxa (Hedberg et al., 2022). Interestingly, they were not particularly functionally unique, which suggests that although they may have possessed distinct functional traits compared to the community at large, many taxa were somewhat functionally redundant with each other. Thus, the megafaunal extinction led to a severe reduction in the functional breadth of the community but had little effect on the overall density of species within remaining functional space (Figure 8E). The surviving large mammals, however, were now the remnant occupants of a once much more populated area of functional space, and consequently, their unique functional contribution skyrocketed in the Holocene. Interestingly, many of these species, such as bison and gray wolf, are now considered keystone species in the ecosystems they still occupy (Knapp et al., 1999; Fortin et al., 2005; Ripple and Beschta, 2012; Ratajczak et al., 2022).

Of course, the decline in large-bodied mammals did not end at the terminal Pleistocene. Increases in human population and land use over the last centuries have led to the continued extirpation of megafauna from many parts of the globe, including North America (Cardillo et al., 2005; Estes et al., 2011; Dirzo et al., 2014; Ripple et al., 2019). This includes the extirpation of bison and jaguar from much of their former geographic range (Hornaday, 1887, Flores, 2016, Sanderson et al., 2002). Indeed, in large part due to the inflated functional uniqueness of surviving large mammals,

declines in functional diversity in the historical and modern periods at Hall's Cave were as great if not greater than those following the terminal Pleistocene extinction (Figure 8A–D). Critically, this suggests that modern ecosystems have reduced ecological resiliency resulting from millennia of past biodiversity erosion, and may be more vulnerable to adverse ecological impacts with continued declines. A similar pattern of accelerating functional loss with past or predicted biodiversity decline has been found in marine megafauna (Pimiento et al., 2020a), and for large, slow reproducing species across the tree of life (Carmona et al., 2021).

The functional space lost in the Pleistocene has not gone completely unused. Humans have changed the function of communities by introducing exotic species to novel habitats. This includes many large-bodied herbivores such as zebra, oryx, gazelle and elephants brought into the United States for sport hunting (Butler et al., 2005). Especially for free-ranging populations that become naturalized, exotics may restore portions of lost ecological function. This is particularly true in South America and Australia, where including introduced herbivores to the native assemblage restored 47% to >100% of lost late Pleistocene functional richness, respectively (Lundgren et al., 2020). Similarly, in central Texas, the modern non-native fauna has 'filled in' areas of functional space once occupied by Pleistocene megafauna (Figure 8E). However, restoring functional richness is not always the same as restoring functional composition; including introduced species generated highly distinct trait distributions from both the modern and terminal Pleistocene communities (Hedberg et al., 2022). Nonetheless, these are provocative results. They suggest that the once outlandish idea of introducing functional replacements into ecosystems (i.e., 'rewilding'; Martin, 1992; Donlan et al., 2005, 2006) may indeed be a path forward to help preserve key ecological processes previously provided by extinct megafauna (Zimov, 2005; Root-Bernstein and Svenning, 2016; Bakker and Svenning, 2018; Lundgren et al., 2020).

## Summary and future directions

The extinction of all terrestrial mammals above ~600 kg in the Americas and Australia (Lyons et al., 2004) marked the beginning of a fundamental transition of Earth's ecosystems; a shift from a world dominated by wildlife to one dominated by people and their livestock (Barnosky, 2008; Smith et al., 2016b). Continued study of the consequences of this event is making the enduring legacy of the extinction increasingly evident. As we have demonstrated here, the influence of late Pleistocene megafauna extinction extended to all aspects of the Earth system – the atmosphere, geosphere, hydrosphere and biosphere. Indeed, many contemporary ecosystems still exhibit the effects of this extinction.

Because the rate of biodiversity loss is rapidly increasing (Ceballos et al., 2015, 2017), scientists do not have time to wait for the results of long-term experimental studies before proposing potential mitigation strategies. Moreover, to date most modern studies of the ecosystem function of large-bodied mammals focus on a single system – the influence of African savanna elephants on vegetation structure and biodiversity (Hyvarinen et al., 2021). Other megaherbivores and aspects of ecosystem function are comparatively neglected leading to considerable deficiencies in our understanding of the ecological role of megaherbivores (Hyvarinen et al., 2021). By integrating paleoecology into conservation biology, we can gain unique long-term perspectives relevant to modern conservation efforts (Dietl and Flessa, 2011; Barnosky et al., 2017; Smith, 2021). Future research should continue to examine the responses of survivors to the late Pleistocene

extinctions and work toward integrating results across taxa and ecosystems. A better understanding of the synoptic role of large-bodied mammals can help garner support for conservation efforts and guide future management decisions.

**Open peer review.** To view the open peer review materials for this article, please visit http://doi.org/10.1017/ext.2023.6.

**Acknowledgments.** Many thanks to R.E. Elliott Smith, who provided analytical and computation assistance and to W. Gearty for help with Figure 7. We thank the Smith Lab at UNM for their work on the Hall's Cave site over the years and the Texas Memorial Museum (particularly Drs. Chris Sagebiel and Ernie Lundelius) for access to collections and assistance in locating references, specimens or information about the fossil sites. We would also like to thank Drs. R. Toomey and T. Stafford, Jr. for their foundational work at Hall's Cave, without which much of our work would not be possible, and the Hall family for graciously allowing generations of paleontologists to work at this very important late Quaternary site.

**Author contributions.** All authors contributed substantially to the conceptualization and writing of the manuscript.

**Financial support.** This work was supported by the National Science Foundation Division of Environmental Biology (Grant Numbers 1555525 and 1744223; F.A. Smith PI; S.K. Lyons and S.D. Newsome, co-PIs), and a National Science Foundation Division of Environmental Biology Research Coordination grant (2051255; S.K. Lyons PI; S.A.F. Darroch, C.V. Looy and P.J. Wagner, co-PIs). E.A.E.S. was supported by a National Science Foundation Postdoctoral Research Fellowship in Biology (DBI-1907163), and a Peter Buck Postdoctoral Fellowship from the Smithsonian Institution.

**Competing interest.** The authors declare none.

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
