## [Reviewer Report]

*Comments to Author*: This review summarizes the current state of our understanding of how the aftermath of the megafaunal extinction at the end of the Pleistocene affected the survivors of this extinction and shaped their communities and thus the ecosystems we see today. The manuscript is rich with information, examples, and references for readers to pursue, thus I anticipate it will be of interest to a broad range of readers across multiple disciplines. It suffers, however, from a lack of cohesion within and across sections, areas of redundancy, and inconsistency with respect to the writing style and level of detail presented. Before it is published, I would recommend that certain sections of the manuscript be significantly revised to make them more even, thorough, and up to date. Thus most of my comments are focused on improving the clarity of the narrative and improving the manuscript’s balance and flow via structural and wording suggestions. I have divided my comments up into two sections: Structural suggestions and Writing / content suggestions.

<b>Structural suggestions:</b>

1) Length and level of detail

The manuscript is highly inconsistent when it comes to the length and level of detail provided in the different sections. Some of this is unavoidable and to be expected - certain research areas have received more study in the past than others, and this imbalance is important to point out in a review (see comment below about highlighting and addressing knowledge gaps). But a lot of it is not due to discrepancies in past research effort and could be remedied through expanded discussion of even just the examples provided. For example, many of the shorter sections include paragraphs where each sentence summarizes a different paper with no additional context. Providing a couple more sentences per study about what was done, where it was done, what was found, and why the findings relate back to the theme of the paragraph would vastly improve the paper and even out the length discrepancies between sections. Not all sections suffer from this (see Section 3 for an example of a section that does not have this issue). The manuscript would be improved if the style and level of detail provided in Section 3 was replicated throughout the manuscript.

2) Road map and parallel structure of sections and paragraphs

The manuscript would benefit significantly from a clearer overall roadmap as well as guides for the reader within sections. Specifically, at the beginning of each section, the reader is not told what concepts they are going to learn about in that section and why they are important for understanding the impacts of megafaunal loss. Furthermore, very few links are drawn between sections or between paragraphs within sections, thus it is hard to grasp a common thread that runs throughout (Sections 1.0 and 5.0 provide examples). Indeed, it sometimes reads as if paragraphs and/or sections were written in isolation from one another despite being on similar topics, which then also leads to unnecessary redundancy across sections.

I believe the disjointed reading experience can be remedied through inserting (or expanding) a paragraph or partial paragraph orienting the reader to what is to come in the opening paragraph of each major section. For example: “First we review what is known about the effects of extant herbivores on ecosystems today. Then we extend these inferences into the Pleistocene and compare with what is known about X, Y, or Z. We find that the effects were likely magnified because of X, Y, and Z.” Or, for section 3: “In this section, we document how the process of digestion within …. Can influence earth systems at global scales through changes in the greenhouse gas content of the atmosphere and redistribution of nutrients on Earth’s surface.” Great examples of such guides are present in subsection 3.1 Biogeochemical change and subsection 4.2 Dietary changes. These subsections read much more smoothly than many of the others and thus would be great examples to use when adding similar paragraphs to other sections. Having these roadmaps in only two subsections, however, makes the writing across the manuscript feel uneven.

3) Knowledge gaps that need filling

I would also suggest the authors be more consistent across sections in pointing out what we know but also what we don’t know. While later sections in the manuscript present future research directions, the first sections do not. Evening out the messaging across sections about gaps in our understanding that need to be filled (i.e. fruitful areas for future research) would be helpful. I also believe such sections are important components of any review paper as they make the paper a useful forward-facing tool in addition to providing a review of the current state of the field.

4) Paragraphs that introduce new concepts in the last sentence

There are multiple instances throughout the manuscript where major new concepts are introduced at the end of paragraphs, without much (or any) definition, explanation, or context. For example, in Line 175 there is a sentence about it being clear from these studies that modern ecosystems in the New World have probably never recovered. This is a big new concept, and also a confusing passage since only one study is actually discussed in the paragraph. The topic of recovery (or lack thereof) is worthy of its own paragraph and, if it is going to be introduced, needs more information to support the claim.

Another example is in Line 630, where the concept of ‘mesopredator release’ is raised in the last sentence of the paragraph (with no definition - see next comment).

Another example is in line 677 where the concept of ‘time binning’ is mentioned. I don’t dispute the importance of time binning - quite the contrary - but this sentence is an outlier as there is no discussion of taphonomy elsewhere in the manuscript. This concept needs to be defined and the reader told why it is important. With respect to taphonomic issues, one could argue it would even warrant a separate box similar to the boxes provided for stable isotopes and functional diversity. But if there isn’t space for a discussion of these topics in the manuscript, then I would quite honestly leave the sentence in line 677 out rather than introduce it with no further discussion.

5) Defining concepts the first time they appear and minimizing redundancy

Due to the issues mentioned above with disjunction between paragraphs, there are multiple instances in the paper where concepts are introduced more than once, and/or a concept is not defined the first time it is presented to the reader. In these instances, the concept should be explained and defined the first time it is encountered and then a reference provided back to that definition later in the manuscript when a concept is raised again. Multiple examples are as follows:

The concept of ‘evolutionary anachronisms’ is first introduced in the last paragraph of section 2.1 (Line 234), but minimal definition is provided. It is then re-introduced in section 2.3 on line 286. I would recommend expanding the definition of evolutionary anachronisms when it is first introduced in 2.1, and then linking the example provided in 2.3 back to that concept (e.g. “Plants on the landscape with no dispersers larger enough to disperse their seeds is another example of an evolutionary anachronism today (see section 2.1).”)

The term ‘mammoth steppe’ is first introduced in section 2.1 (line 226) but no definition or geographic context for the term are provided. The second mention of ‘mammoth steppe’ occurs in section 3.1 (line 380), and this time it is parenthetically defined and geographic context given. The definition provided in line 380 needs to be moved up much earlier when the concept is first introduced.

The term ‘ecological release’ is introduced on line 523 but not defined. It needs a definition.

The concept of ‘environmental engineering’ is raised on line 164 but not defined. It is also not clear how the authors distinguish between environmental engineering and the concept of ‘ecological engineering,’ which is discussed elsewhere several times (sections 2.2 and 5.1). The authors should clarify if these are actually two different concepts, or streamline if they are using two separate terms for the same concept. If the latter, I would recommend picking and using just ‘ecological engineer/engineering.’

The concept of ‘ecological engineers’ is first introduced to set the stage for what is to come in section 2.2, but then the concept is actually not discussed in the subsequent paragraphs of this section. Thus the paragraphs feel like they lack a clear thread. If ecological engineering is the theme that is going to be used to introduce this section, then the examples given in the following paragraphs should also be explicitly tied back to it. The concept of ‘ecosystem engineers’ is brought up again in section 5.1 (line 708). But again it is not discussed or defined.

The term ‘mesopredator release’ is introduced on line 630 but not defined. It needs a definition.

6) Passive vs. active voice

The sections of the manuscript are inconsistent in the style of writing used to present studies. For example, Section 2 is primarily written in a passive voice that highlights the research instead of the researchers, while Section 3 primarily uses an active voice that highlights the researchers (e.g. “X et al. did Y and found Z”). While either approach is ok to use, the authors should be consistent. My personal preference would be to align all sections via active voice, similar to the writing in Section 3. I feel use of active voice would also provide more opportunities to seamlessly work in missing context (e.g. where a study was done or methodological details on how it was done), as well as provide opportunities to add in more support for the conclusions that the authors draw from their review.

7) Focus on Hall’s Cave

Many of the examples provided in the manuscript, especially in the later sections, are from recent work done at Hall’s Cave in central Texas. This is a spectacular site and the examples of insights learned from analysis of this record that are included are excellent. But I wonder if more explicitly introducing it as a case study in the abstract and/or introduction would be good so that the reader is expecting that focus. Instead, if feels like the authors have to re-introduce the site multiple times throughout the manuscript, leading to some unnecessary redundancy. For example, something could be included about the special focus on Hall’s Cave in the paragraph beginning on line 130, which lays out the goals for the paper.

8) Some thoughts on causality

Throughout the manuscript there are issues with causality; paragraphs often include statements assuming that the changes observed in species or communities or ecosystems are directly due to the loss of megafauna / biodiversity. I am not arguing these statements are unfounded - I know the authors have worked on issues of causality extensively. But the statements in the manuscript implying causality need more support and/or information on how causality has been established in the studies that are being reviewed. As they stand right now, some of the points come across as “just believe me” statements, which I am sure is not the goal. So I suggest a re-working of these areas to provide the reader with more information about why causality is likely vs. potential effects of other changes that were going on at the time, for example in the climate system. It would also be beneficial to be more explicit about whether a response is directly related to the loss of megafaunal biodiversity or is a response to the myriad of cascading but indirect ecosystem effects that the extinction triggered.

<b>Writing and content suggestions</b>

Abstract on cover page says “top consumers” but abstract in the text (page 2) says “large-bodied herbivores.” I assume the second is correct.

Keywords list allometry and lateral transport, but those terms and concept don’t come up later in the manuscript. Suggest removing or swapping for some more closely aligned terms (e.g. Hall’s Cave, Isotope Ecology, Pleistocene…).

1.0 Movement

Typo in line 146

Line 150: the discussion of slope angle is interesting but the purpose of it is not entirely clear. I thought the paragraph was headed towards the fact that changes in water movement must impact landscape erosion. But instead, I think the goal was to highlight the high energetic costs associated with being big. I recommend clarifying the goal by adding a thesis statement at the beginning of the paragraph and then making sure to link that concept with the following paragraphs.

Line 161 typo: should be “Gabon’s forests”

Line 170: not sure what “their” is referring to. I think it’s megafauna, but it could also refer to the physiological effects of the megafauna. Should be clarified.

2.0 Foraging

Line 181: Recommend adding some sort of link to the previous section, such as “As with movement, the act of foraging and the food they consume…”

Sentence covering line 182 - 184 needs clarification/wordsmithing. It’s a long list, and as written, it is not clear if the authors are saying that the entire list is contributing to changes in albedo or just soil disturbance.

2.1 Vegetation structure and composition

The sentence that begins on line 197 would benefit from multiple commas.

Line 206: Needs more context, especially a time frame reference. e.g. “…foraged across the entire carbon isotope spectrum during the Pleistocene.”

The sentence starting on line 208 doesn’t fit at the end of this paragraph. I recommend moving it up into the first paragraph of section 2.0, where there is already a similar list, which would also reduce redundancy. e.g. if moved to line 181, it could read “…has important ecosystem consequences that vary along rainfall and soil fertility gradients.”

Line 219-221, consider streamlining to: “…grazing would have increased the quality of graze in the ecosystems where they occurred, facilitating coexistence of…” As an aside, wouldn’t the increased quality / nutritional / caloric content also facilitate support of larger population sizes than today? Not sure if there are any studies supporting this but that would be an interesting additional topic to mention here.

Summary of Dantas and Pausas (2020) is an example of a passage that would benefit from being re-written in active voice and from inclusion of more context (e.g. lead with its geographic context).

2.2 Foraging and Species Diversity

This section could use a more informative title. Perhaps something like “Diversity of resource and consumer communities.” It also stands out because the other subsections in the Foraging section don’t lead with “Foraging and…”

Line 248: Unclear if “other species” is referring to an increase in other plant species or other consumer species, potentially through an increase in graze quality (which was discussed in the preceding subsection). Need to clarify who was facilitated and how.

Line 249: define ‘wallows.’

Line 251: This sentence needs context with regards to spatial scale of the study and the documented effects.

Line 251: Because soil compaction also happens with movement, this content could also fit earlier. To clarify that it belongs here, I suggest changing “herbivore activities” to “herbivore foraging.”

Line 261: Richness and abundance of the smaller species? I think so, but need to clarify.

Line 265: Not sure what the “However” at the beginning of the sentence is contrasting with. Recommend removing.

Line 266: What is meant by low-level coverage? Shrubs? Clarify.

Lines 269-270: Louthan sentence is trying to fit a lot of new concepts into one sentence right at the end of a paragraph. It is also not grammatically correct. It needs to be either re-worked to tie these new concepts / new system (plants-pollinators) back into the rest of the paragraph / section or leave it out.

2.3 Seed dispersal

Line 283: Should explain why passage through the digestive system is required for germination.

2.4 Fire regimes

Line 296: Typo. Sentence needs an “of.”

Line 298: An example of a place where flow could be improved with some simple wording changes. For example, adding “As discussed above, herbivore foraging alters the abundance of……, which in turn influences the frequency of…” .

Line 302: Another suggestion to improve flow: begin sentence with “Given these effects, promotion of grazing… has recently been proposed as…” These types of additions will help tie content together.

Line 311: Rewrite in active voice: “In an analysis of sedimentary charcoal records from XXX, Dana et al. 2012 detected a widespread increase….”. This example also currently lacks geographic context. Is it from Australia like the following sentence? Should clarify this.

3.0 Digestion

Line 323: I would suggest flipping this sentence to clarify that the time frame of one day holds for both kg of vegetation consumed and hours spent feeding. “…a wild adult elephant can feed up to 18 hrs per day and can consume as much as 200-300kg of vegetation in that time-frame.”

3.1 Biochemical change

Line 337: suggest adding ‘methane' between atmospheric and concentrations

3.2 Redistribution of nutrients

Line 409: The sentence states: “…elephant home range can reportedly exceed 8,500 to 10,000km2 (Lindeque and Lindeque 1991, Ngene et al. 2017).” In section 1.0 on line 154, however, it states that “…elephants can travel more than 2800 km annually (Mills et al. 2019).” These are very different estimates, have different units, and come from different sources. So… which estimate is it? Or how do these differ? These should be reconciled or clarified so the manuscript is presenting consistent information across the sections.

The paragraph starting on Line 453 is about foraging and movement altering surface albedo and fire frequency. Impacts to surface albedo from foraging and movement are also presented in section 2. To minimize redundancy and ensure consistency, I would recommend condensing this paragraph into section 2 and potentially referring back to it here rather than re-presenting it as a new concept.

4.0 Surviving mammals

Paragraph starting on line 466: this is great in that it sets up areas of knowledge gaps that need more study. But it falls short in that it offers no summary of the subsections to come. Specifically, this paragraph is focused on energy flow, but the subsection that follows does not mention energy flow, it is about geographic distributions and abundance. While there are connections between abundance and distribution and energy flow, these are not explicitly laid out for the reader. Thus the intro paragraph feels disconnected from the rest of the content of the section.

4.1 Distributional / Abundance changes - why the slash in the title? Suggest changing it to “Changes in Distribution and Abundance.”

Line 494: suggest inserting “… result from time averaging or stratigraphic mixing, these factors seem unlikely to explain.…”

Line 496: I would recommend removing ‘however’ and using this sentence to start the next paragraph. This is because it introduces the idea that other non-climatic factors were also important. For example, start the next paragraph with: “That species shifted their ranges… suggests other factors in addition to climate were important. But the extent to which shifts in mammalian geographic ranges driven by biodiversity loss remains an open question. Niche modeling shows…”

Line 515: recommend switching to active voice for this entire paragraph.

Sentence that starts on 526, which is currently the last sentence of the paragraph, is a good sentence but reads more like a thesis statement that would lead a paragraph, not end it. Thus I would suggest using it earlier to lead off a paragraph.

Line 531: It also seems like there could have been an increase in abundance due to exploitation of non-food resources (e.g. territory, breeding sites, etc.). It’s unclear to me why the focus here is restricted to just food resources. I would recommend clarifying why food, if it really is a point that is specific only to food resources, or removing ‘food’ to leave it broad at just ‘resources.’

Line 533: - “…which homeostasis theory predicts would lead to reorganization of surviving lineages.” Meaning what? This isn’t specific to abundance, as far as I know. This point needs expansion and clarification if it is important to include.

Line 534: Abundance is definitely difficult to characterize in the fossil record, but readers might not know why. I would suggest adding a sentence or two about what makes it hard. Also, this paragraph is about the occurrence and frequency of bison herds across the landscape, not abundance of bison, per se. While the two obviously go hand in hand, it is important to keep the units clear, so I would suggest revising the paragraph accordingly. (e.g. “… shifts in both distribution and abundance of bison herds to tracking of favorable environmental conditions, they do note a spike in herd abundance at the…”

4.2 Dietary changes

Lines 561 - 563: This background information is redundant with several other areas in the paper. Suggest removing or condensing elsewhere.

Line 564: This list of survivors could fit well if moved up to a revised intro paragraph for section 4.0.

Line 580: the reference to figure 6 here seems inappropriate and should be removed. Figure 6 doesn’t show any large herbivores. Instead, it shows rabbit and rat sized herbivores. Figure 6 is more appropriately referenced in the next paragraph (Line 594).

Line 594: a great place to switch to active voice. “Tome et al. XXX showed that the d13C values of S. Hispidus…. However, in a different study, Tome et al. 2022 showed that isotopic values of Neotoma.… Meanwhile, Smith et al. 2022 showed….”

Line 601: I would recommend adding “…complex responses by animals to the multiple effects of the megafaunal extinction…” because the responses of the small mammals aren’t necessarily directly to the extinction itself, but to the ecological and environmental cascades that followed.

Box 1 - Stable Isotopes

This box provides a very nice and concise explanation of how stable isotopes are used in ecology and paleoecology; specifically the influence of photosynthetic pathway and trophic position on d13C and d15N values, respectively. It makes no mention, however, of the climate sensitivity of these isotope systems. While secondary in importance and, likely, magnitude of effect, climate (e.g. changes in aridity) can drive isotopic signals in the resource base which then are incorporated into the tissues of consumers. This is an important effect to account for when comparing dietary shifts within a species across time periods where climate is also changing, thus it deserves mention in this paragraph. Thus I recommend adding in mention of the climatic sensitivity of these isotopic systems and some information on how one would expect enrichment to change under scenarios of warming/drying and/or cooling/getting wetter.

4.3 Ecomorphological changes

While the header of this section refers to ecomorphological changes, the section actually focuses solely on body size. Thus it might be good to re-title this section as Body size changes instead.

The section would benefit from information on why body size is an important trait for analysis (arguably the most important!). As it currently stands, there’s no justification given for looking at body size, and no information provided on why it is important (e.g. scaling of vital rates in mammals, impacts on species interactions, etc.). If this section could be expanded with respect to background such that it is more similar to the level of detail that was presented in the isotope section preceding it, it would be highly beneficial to the manuscript.

The literature cited in the first paragraph is fairly old; much has been done on body size responses to climate change since 2006, including by the authors. I suggest updating this paragraph with a summary that includes more modern references.

Line 651: the last two sentences of this paragraph stood out as an odd fit and even perhaps somewhat contradictory with the results presented in section 4.2. The sentence as it stands reads “Were changes in diet reflected in shifts in body size or other aspects of morphology like tooth structure?” This is a fine question except that the preceding section documents inconsistent and often no changes in diet via isotope analysis. Furthermore, the rest of section 4.3 does not discuss tooth structure, or any other ecomorphological traits beyond body size. I suggest revising the question at the end of this paragraph to better align with the points made in the previous section, and to be specific to body size (not tooth morphology) to better reflect the content of this section.

Line 660: I recommend using the sentence to tie ideas back to the earlier discussion of range shifts by wordsmithing as follows: “As with range shifts, the shifts in body size of species from the P to the H were also species specific, with some animals becoming significantly larger (e.g. deer, jackrabbits), others smaller (e.g. bison, cottontails) and others exhibiting no change (e.g. carnivores).”

Line 663: I would argue that it is more that species (not ‘animals,’ line 662) responded in a multitude of ways (not just ‘other’ ways), and would recommend referencing other relevant sections in this paper that support that (e.g. abundance and distribution section).

Line 665: “community changes" is vague. What is actually meant by this? Structural changes? Compositional changes? Should clarify.

Line 665: This paragraph got me to wondering if body size responses at Hall’s Cave were patterned across types of distribution or abundance changes. Not sure if this question has been asked before but it seems like the record at Hall’s Cave might be up to addressing it. Could be used as an example of an “open question” type sentence to insert here (e.g. “Future research should address whether…”)

Line 666: I would suggest removing the “biodiversity loss led to larger body size” statement, unless more information is included about how that causality is established.

Line 671: Neotoma and Sigmodon should be referred to as genera, not species.

Line 674: Suggest inserting “These individualistic body size responses…”

Line 675: Suggest removing the first part of the sentence and rewording as “Future work needs to examine responses (what does responses mean? Body size trajectories? Clarify) across a spectrum of body sizes, habitat preferences, and dietary strategies.”

5.0 Changes in communities and ecosystems

This section would benefit from an overview / roadmap in the first paragraph

5.1 Ecological Interactions within communities

Line 698: is it actually a full on collapse? Or is it a (major) reorganization? One could argue that it was more along the lines of reorganization since the entire ecosystem didn’t collapse. If collapse is the correct word, then by all means use it. But a bit more explanation of what is meant by collapse / justification of the use of collapse would be beneficial.

Line 701: what is meant by “robust food web”? Defined or measured how?

Line 709: this paragraph lacks reference to geographic location until the very last sentence, which is confusing for the reader. Suggest rewording the beginning to “…an in-depth analysis at Hall’s Cave in central Texas that used stable isotopes to track dietary niche shifts found that…”

Line 711: I am confused by this contrast: “Surviving canids did not colonize the niche previously occupied by the dire wolf.” But then, in the next sentence, “…the shift of a former mesopredator into the apex predator role…” These seem at odds with one another so need to be clarified. Which mesopredator are you talking about?

Line 725: the sentence ends with “… when considering only mammals larger than 1 or 5 kg.” Which size cut off is important? Both? And if both, was there a difference between the size thresholds? If so, what was it? This sentence needs clarification, otherwise it’s not apparent what these thresholds mean or why they are presented.

Line 720: This paragraph on homogenization and range expansions seems like it would have a better home earlier in the manuscript in the section about geographic range responses. It does not currently have clearly defined links to species interactions, which is the focus of this section.

Line 720 (cont): Also, Section 4.1 talks about how geographic range responses were individualistic. But here, in lines 726-730, it is stated that surviving mammals expanded their geographic ranges. These two paragraphs are not consistent in the message they are presenting about geographic range responses, thus should be reconciled. Of course, it is entirely possible for there to be an overall pattern of expansion but range responses still quite individualistic. But if that is indeed the case, it needs more explanation and discussion, ideally with that information all contained in the same paragraph.

Line 730: the last sentence here could use more explanation. What does this point mean? What are the implications?

5.2 Functional changes within communities

The set-up and introduction of concepts in this section is much smoother and more detailed than other sections in the manuscript, thus would be an excellent guide to use for improving the other sections and making the manuscript more cohesive.

Line 852: the statement that similar patterns have been found elsewhere needs more detail. Among what other taxonomic groups and on what continents / environments?

Line 866: This sentence Introduces the concept of rewilding - again at the end of a paragraph - but does not offer a definition of the concept. I don’t think one can assume that the reader is necessarily familiar with this concept, thus a short definition / explanation should be included, along with the references.

References

Martinzes-Meyer et al. (2004) is included twice

Figures and captions

Fig 1: The y axis of the figure reads % Extinction intensity. But the unit is the proportion of taxa that went extinct, right? That is what the figure caption says. I recommend updating the y axis to match the figure caption (i.e. remove “intensity”).

Fig 3 caption, line 1541: Typo: should be “trail” not “trial”

Figure 5 is currently mislabeled as Figure 6.

Fig 5: Species names should be italicized.

Figure 6 is mislabeled as figure 7.

Fig 6 caption, line 1565: Suggest adding in the specific tissue used: “…bulk d13C and d15N values from bone collagen of mammalian survivors.”

Also - I’m pretty sure that stat_ellipse draws a data ellipse and not a confidence ellipse. Or at least it used to (perhaps the package has been updated?). But if it is still drawing data ellipses, it is not representing a true 95% confidence interval, and the ellipses aren’t sensitive to group sample size. I can’t infer sample sizes from this plot, and whether the ellipses are sensitive to sample sizes, since no N values or data points are shown. Given that the sample sizes of different groups are almost certainly not the same (just due to the nature of working with these types of data), and that some ellipse methods can be quite sensitive to sample size, I would suggest that at least the sample sizes for the groups should be reported or the actual data points shown within the ellipses. It would also be good to describe if and how the issue of unequal sample sizes was accounted for in the analysis.

Fig 6: Canis latrans silhouette has a big black box over it. Something isn’t rendering in the pdf correctly.

Figure 7 has no figure label.

Figure 8 is mislabeled as Fig 9

---

## [Reviewer Report]

*Comments to Author*: This is a great review about the ecological consequences of late Pleistocene mammalian extinctions. I have a few suggestions to improve readability and additional references that could enrich the discussion.

Ln 74. This sentence is a bit confusing because, as explained later, the late Pleistocene extinction is not formally considered a mass extinction event.

Ln 261-270. This paragraph about the effects of excluding large-bodied mammals is a bit hard to follow. For instance, it is not clear how the absence of herbivores altered vegetation structure. The explanation about the effects over pollinators is also a bit superficial.

Ln 293. It would be useful to say that there is little evidence of plants extinctions as a result of seed dispersal limitation. Some authors suggest several species may have been rescued by human use.

Ln 312. These results about the vegetation shifts in Australia and fire frequency have been explored in previous paragraphs.

Ln 467. Figure 1 has no information about energy flow. I suggest moving the citation.

Ln 471. It seems Barnosky et al. 2008 should be cited here.

Ln 531. This sentence sounds finalist

Ln 532. The concept of vacant niches is controversial. I suggest rethinking these sentences.

Ln 629. It would be helpful to cite Hayward et al (2016; doi: 10.3389/fevo.2015.00148) about dietary shifts in P. onca.

Ln 702. I suggest citing Pires et al (2015; doi: 10.1098/rspb.2015.1367) here, as it discusses the consequences of structural changes in ecological networks since the Pleistocene.

Ln 742. I suggest avoiding to call co-occurrence patterns interaction strength as it can be misleading.

Ln 789-796. Lundgren et al (2020;doi: 10.1073/pnas.1915769117) should be cited here

Ln 880-890. A few papers that could be cited here:

Dietl et al (2011; doi:10.1016/j.tree.2010.09.010);

Barnosky et al. (2017; doi: 10.1126/science.aah4787)

Ln 1541. Typo: trial

Figure 5. The entire text focuses on large mammals. This figure about small mammal redistribution seems out of context.

Figure 7. there is something wrong with the silhouettes

---

## [Reviewer Report]

*Comments to Author*: I enjoyed reading the paper. It is thorough and informative and certainly deserves to be diffused to the community. Point is, I have had a hard time understanding whether it is addressing the North American or worldwide legacy of late Pleistocene extinctions. In some passages, the manuscript truly focuses on the former. An emblematic example at line 380:381, where the authors write:

The wide-scale habitat alteration from the vast ‘mammoth steppe’ of the Pleistocene (a diverse assemblage of grasses, forbs and sedges covering much of North America)

but there was mammoth steppe over most of North Eurasia as well!:

R. Dale Guthrie, Origin and causes of the mammoth steppe: A story of cloud cover, woolly mammal tooth pits, buckles, and inside-out Beringia. Quat. Sci. Rev. 20, 549–574 (2001).

A second issue regards the quite overlooked effects of indirect (including apparent) competition (e.g. lines 118, 242, 385, 504-510, and others).

Megaherbivores are more than just ecosystem engineers, removing megaherbivores relaxes predation on predator-limited smaller prey and increases the resource base for predators. It has been demonstrated that this may have long-term consequences on species' survival. And overgrazing megaherbivores may disfavor mesoherbivores, the converse at high density. The authors should at the very least acknowledge these effects and cite (the list is much longer but this is sort of minimal requirement):

W. J. Ripple, T. M. Newsome, C. Wolf, R. Dirzo, K. T. Everatt, M. Galetti, M. W. Hayward, G. I. H. Kerley, T. Levi, P. A. Lindsey, D. W. Macdonald, Y. Malhi, L. E. Painter, C. J. Sandom, J. Terborgh, B. Van Valkenburgh, Collapse of the world’s largest herbivores. Sci. Adv. 1 (2015), doi:10.1126/sciadv.1400103.

P. Chesson, J. J. Kuang, The interaction between predation and competition. Nature. 456, 235–238 (2008).

and even direct evidence on Pleistocene megafauna:

C. Meloro, P. Raia, C. Barbera, Effect of predation on prey abundance and survival in Plio-Pleistocene mammalian communities. Evol. Ecol. Res. 9, 505–525 (2007).

minor points:

line 84. there is a good amount of work done on late Quaternary/holocene extinctions, especially on islands, that is worth mentioning and goes beyond Turvery's account:

birds:

1. A. Fromm, S. Meiri, J. McGuire, Big, flightless, insular and dead: Characterising the extinct birds of the Quaternary. J. Biogeogr. 48, 2350–2359 (2021).

reptiles:

A. Slavenko, O. J. S. O. J. S. Tallowin, Y. Itescu, P. Raia, S. Meiri, Late Quaternary reptile extinctions: size matters, insularity dominates. Glob. Ecol. Biogeogr. 25, 1308–1320 (2016).

line 221. it might be important to cite indirect effects on vegetation structure via trophic cascades here:

M. L. Pace, J. J. Cole, S. R. Carpenter, J. F. Kitchell, Trophic cascades revealed in diverse ecosystems. Trends Ecol. Evol. 14, 483–488 (1999).

J. G. C. Hopcraft, H. Olff, A. R. E. Sinclair, Herbivores, resources and risks: alternating regulation along primary environmental gradients in savannas. Trends Ecol. Evol. 25, 119–128 (2010).

In sum, I feel a more balanced selection of papers reporting evidence other than North America can only be beneficial to this otherwise fleshy, robust paper.

regards

---

## [Editor Report]

*Comments to Author*: We want to thank you for your well received review of the late Pleistocene megafaunal extinctions. The three collegial reviews have all recommended minor revisions to the manuscript. Reviewers 1 and 2 are complimentary of the scope and depth of the review, but do recommend some clarifications and additions that we would like you to carefully consider. Reviewer 3 is also complimentary about the amount of useful information in the manuscript. They suggest more substantial changes to the manuscript that are divided into structural changes and writing/content changes. We found that all of the reviews provide excellent suggestions that will ultimately improve the appeal of your excellent contribution. Again, thank you for your submission and please carefully consider and incorporate the recommendations of the reviewers, which have given very constructive feedback.

---

## [Reviewer Report]

*Comments to Author*: Dear Dr Smith,

I'm glad to say the ms scope and goal is clear now and wish to strongly recommend the publication of this fine contribution.

regards

Pas

---

## [Reviewer Report]

*Comments to Author*: I reviewed the previous version of this manuscript and I consider the authors responded satisfactorily to my comments and to those of other reviewers. The revised version has improved considerably, and I think this is a relevant contribution to the field. I have only a few minor suggestions:

Page 3, line 4. I suggest rewriting this sentence. The word exclusively makes it confusing

Pg 20. "Because the passage rate of food through the body is positively associated with body size (Peters 1983)". Is this sentence correct? Retention times increase with body mass.

Pg 40. The second sentence seems to contradict the previous sentence

---

## [Editor Report]

*Comments to Author*: Dear Dr. Smith and co-authors,

Thank you for your thoughtful response to the reviewers suggestions and improvements to your manuscript. This looks to be an excellent contribution to the journal. One of the reviewers had a few minor suggested changes that you should consider prior to finalizing the manuscript. Following consideration of those changes and those that I list below, we will accept your manuscript for publication. 

Sincerely,

Greg Wilson Mantilla

Minor fixes:

1) p. 22, 2nd paragraph, "effected" should be changed to "affected"

2) same line, Late Pleistocene should be late Pleistocene (other instances have Terminal Pleistocene and should be terminal Pleistocene)

3) p. 24, line 5 "Neotoma sp." should only have "Neotoma" in italics not "sp."

4) p. 25, 2nd to last line, (e.g., Chesson and Kuang 2008 requires a complete pair of parentheses.

5) p. 26, line 5, no comma after North

6) p. 29, 2nd paragraph, should be a comma after maximum prey size and only 2016 should be in parentheses (not Van Valkenburgh et al.)

7) p. 36, 1st paragraph, line 11, there is a double occurrences of "in" 

8) next line, remove the comma after monocultures it is not needed

9) p. 38, Terminal should be lowercase

10) p. 41, Modern is capitalized here but not elsewhere. Probably should be capitalized throughout